

**Responses of field-grown maize to different soil types, water regimes, and**
**contrasting vapor pressure deficit**
Thuy Huu Nguyen[1, *], Thomas Gaiser[1], Jan Vanderborght[3], Andrea Schnepf[3], Felix Bauer[3], Anja
Klotzsche[3], Lena Lärm[3], Hubert Hüging[1], Frank Ewert[1, 2]
[1]University of Bonn, Institute of Crop Science and Resource Conservation (INRES), Katzenburgweg 5, 53115
Bonn, Germany
[2]Leibniz Centre for Agricultural Landscape Research (ZALF), Institute of Landscape Systems Analysis,
Eberswalder Strasse 84, 15374 Muencheberg, Germany
[3]Agrosphere (IBG-3), Institute of Bio- and Geosciences, Forschungszentrum Jülich GmbH, 52428, Jülich,
Germany
*Corresponding author, email: tngu@uni-bonn.de
**Abstracts**
Drought is a serious constraint to crop growth and production of important staple crops such as maize.
Improved understanding of the responses of crops to drought can be incorporated into cropping system
models to support crop breeding, varietal selection and management decisions for minimizing negative
impacts. We investigate the impacts of different soil types (stony and silty) and water regimes (irrigated
and rainfed) on hydraulic linkages between soil and plant, as well as root: shoot growth characteristics.
Our analysis is based on a comprehensive dataset measured along the soil-plant-atmosphere pathway at
field scale in two growing seasons (2017, 2018) with contrasting climatic conditions (low and high VPD).
Roots were observed mostly in the topsoil (10-20 cm) of the stony soil while more roots were found in the
subsoil (60-80 cm) of the silty soil. The difference in root length was pronounced at silking and harvest
between the soil types. Total root length was 2.5 - 6 times higher in the silty soil compared to the stony



soil with the same water treatment. At silking time, the ratios of root length to shoot biomass in the rainfed
plot of the silty soil (F2P2) were 3 times higher than those in the irrigated silty soil (F2P3) while the ratio
was similar for two water treatments in the stony soil. With the same water treatment, the ratios of root
length to shoot biomass of silty soil was higher than stony soil. The observed minimum leaf water potential
($\psi_{leaf}$) varied from around -1.5 MPa in the rainfed plot in 2017 to around -2.5 MPa in the same plot of the
stony soil in 2018. In the rainfed plot, the mimimum $\psi_{leaf}$ in the stony soil was lower than in silty soil from
-2 to -1.5 MPa in 2017, respectively while these were from -2.5 to -2 MPa in 2018, respectively. Leaf water
potential, water potential gradients from soil to plant roots, plant hydraulic conductance ($K_{soil\_plant}$),
stomatal conductance, transpiration, and photosynthesis were considerably modulated by the soil water
content and the conductivity of the rhizosphere. When the stony soil and silt soil are compared, the higher
'stress' due to the lower water availability in the stony soil resulted in less roots with a higher root tissue
conductance in the soil with more stress. When comparing the rainfed with the irrigated plot in the silty
soil, the higher stress in the rainfed soil resulted in more roots with a lower root tissue conductance in the
treatment with more stress. This illustrates that the 'response' to stress can be completely opposite
depending on conditions or treatments that lead to the differences in stress that are compared. To respond
to water deficit, maize had higher water uptake rate per unit root length and higher root segment
conductance in the stony soil than in the silty soil, while the crop reduced transpired water via reduced
aboveground plant size. Future improvements of soil-crop models in simulating gas exchange and crop
growth should further emphasize the role of soil textures on stomatal function, dynamic root growth, and
plant hydraulic system together with aboveground leaf area adjustments.
**Key words:** irrigation, plant hydraulic conductance, transpiration, root length, soil types, soil to leaf water
potential, stomatal regulation
**Abbreviations:** DOY: day of the year; DAS: day after sowing; TUE: transpiration use efficiency; SF: sap flow;
LAI: green leaf area index; PAR: photosynthetically active radiation; VPD: vapor pressure deficit; An: net





leaf photosynthesis; E: leaf transpiration; $\psi_{leaf}$: leaf water potential; $\psi_{sunlitleaf}$: leaf water potential of sunlit
leaf; $\psi_{shadedleaf}$: leaf water potential of shaded leaf; $K_{soil}$: hydraulic conductance of soil; $K_{root}$: root hydraulic
conductance; $K_{stem}$: stem hydraulic conductance; $\psi_{soil\_effec}$: effective soil water potential; $\psi_{difference}$:
difference between effective soil water potential and sunlit leaf water potential; $K_{soil\_root}$: root system
hydraulic conductance (includes soil and root hydraulic conductance); $K_{soil\_plant}$: whole plant hydraulic
conductance (includes below and aboveground components).
**1. Introduction**
Maize (*Zea mays L.*) is a major staple crop throughout the world. Drought stress, which negatively affects
crop growth and yield, is of increasing concern in several important maize cultivating regions (Daryanto et
al., 2016). Increases in frequency and severity of drought events due to climate change have been recently
reported (IPCC, 2022). Thus, field observations and understanding on how maize responds to water stress
are necessary to suggest promising traits for breeding programs (Vadez et al., 2021) as well as irrigation
schemes (Fang and Su, 2019; Q. Cai et al., 2017). Improved understanding of crops' response to drought
can be incorporated into soil-crop models (e.g. crop modelling and soil-vegetation-atmosphere transfer
modelling).
Stomatal regulation is often considered as a key aboveground hydraulic variable in regulating water use
of crops. Maize was considered as isohydric plant in which stomata are closed in response to sensing
drought conditions to maintain leaf water potential ($\psi_{leaf}$) above critical levels ($\psi_{threshold}$ or minimum $\psi_{leaf}$)
(Tardieu and Simonneau, 1998). Investigations of how stomatal controls differ among species and
genotypes commonly observed minimum $\psi_{leaf}$ or analyzed of genetic variability of stomatal control in
response to varying soil water content. Analyzing measurements of $\psi_{leaf}$ from 400 lines of maize of tropical
and European origins under greenhouse and growth chamber conditions, Welcker et al. (2011) reported
values of minimum $\psi_{leaf}$ from -0.8 to -1.5 MPa, indicating genetic variability of stomatal responses. The
isohydric behavior is due to different mechanisms including hydraulic and/or chemical (e.g. abscisic acid



[ABA]) signals (Tardieu, 2016). The degree to which these underlying mechanisms interact and differ
among genotypes and/or environmental scenarios in explaining the stomatal regulation is still debated
(Tardieu, 2016, Hochberg et al., 2018). Field evidence in variation of the minimum $\psi_{leaf}$ of maize due to soil
water availability is rarely reported.
Water flow along the soil-plant-atmosphere continuum is determined by a series of hydraulic
conductivities and gradients in water potential. Hydraulic conductance of soil ($K_{soil}$), root hydraulic
conductance ($K_{root}$), and stem hydraulic conductance ($K_{stem}$) determine water potential from soil to root
and root xylem water, and thus magnitude of $\psi_{leaf}$. There are two main resistances to water flow from the
soil to the shoot, namely the soil and the root resistances, often expressed as their inverse, $K_{soil}$ and $K_{root}$
(Nguyen et al., 2020; Cai et al., 2018). In wet soils, the soil hydraulic conductivity is much higher than that
of roots, and water flow is mainly controlled by root hydraulic conductivity (Hopmans and Bristow, 2002;
Draye et al., 2010). It is well-known that a decrease in soil matric potential and soil hydraulic conductivity
triggers stomatal closure and thus results in reduction in transpiration rate (Sinclair and Ludlow, 1986;
Carminati and Javaux 2020; Abdalla et al., 2021). For the root water uptake and controlling stomata, the
location where soil and roots are in close contact (rhizosphere) is most important, because when this thin
layer of rhizosphere is disconnected (i.e. soil-root contact is lost), the water movement from soil toward
the roots is reduced, which might trigger stomatal closure to maintain hydraulic integrity of plant
(Carminati et al., 2016; Rodriguez-Dominguez and Brodribb, 2019; Abdalla et al., 2022). The magnitude of
the drop of water potential between bulk soil and soil-root interface increases considerably at different
levels of soil dryness for different soil types (Carminati and Javaux, 2020; Abdalla et al., 2022). Hydraulic
limits in the soil (Carminati and Javaux, 2020), or in the root–soil interface [as measured for olive trees by
Rodriguez-Dominguez and Brodribb, 2019 or tomato (Abdalla et al., 2022)], or in the root properties
(Bourbia et al., 2021; Cai et al., 2022; Nguyen et al., 2020; Cai et al., 2018) or due to both soil textures and
root phenotypes (Cai et al., 2022b) emphasized the importance of belowground hydraulics (Carminati and





Javaux, 2020). However, also the shoot hydraulic conductance could be limiting in some crop plants
(Gallardo et al., 1996) or in trees (Domec and Pruyn, 2008; Tsuda and Tyree, 1997). Stomatal conductance
and shoot hydraulic conductance showed close links to each other in pine trees (Hubbard et al., 2001).
This summary illustrates three points: (i) current studies have often focused either on above or on below
hydraulic limits, but rarely consider both (ii) it is unclear the roles and relations of soil hydraulic properties
to root and plant hydraulic conductance (thus influences on stomatal conductance) (iii) the role of different
hydraulic processes across the soil - plant - atmosphere continuum i.e. soil to roots, stem, and soil-plant
hydraulic conductance in controlling stomatal conductance remains unclear.
Simultaneous measurements of atmospheric conditions (light intensity and vapor pressure deficit), leaf
water potential, and transpiration rates, coupled with measurements of root, stem and whole soil-plant
hydraulic conductance, root architecture, and soil water potential distribution could reveal the relative
importance of rhizosphere, shoot and root growth, and hydraulic conductance vulnerability, especially
under progressive soil drying at field conditions (Carminati and Javaux, 2020; Tardieu et al., 2017). For the
soil water conditions, soil texture and hydraulic characteristics are very important that influence soil water
movement and thus affect infiltration, surface and sub-surface runoff, and ultimately plant available soil
water (Vereecken et al., 2016). Soil texture properties, characterized by different fractions of clay, silt, and
sand particles, are important drivers in determining the soil water retention properties (Scharwies and
Dinneny, 2019; Stadler et al., 2015; Zhuang et al., 2001). Soil with higher water holding capacity (here the
silty soil with low stone content) have a larger amount of plant available water which in turn enables crops
to better meet the evaporative demand and facilitates better crop growth as compared to the soil with
high stone content (Nguyen et al., 2020; Cai et al., 2018). Estimations of hydraulic conductance (different
organs and whole plant hydraulic conductance) were done for crop plants and maize mainly under
controlled environment or pot conditions e.g. for different species and genotypesduring soil drying (Sunita
et al., 2014; Choudhary and Sinclair, 2014; Abdalla et al., 2022; Meunier et al., 2018; Wang et al., 2017; Li



et al., 2016) or various species and genotypes together with different soil textures (Cai et al., 2022a), or
soil texture with different vapor pressure deficit (VPD) (Cai et al., 2022b). Compared to the substantial
effect of soil texture, there was no evidence of an effect of VPD on both soil–plant hydraulic conductance
and on the relation between canopy stomatal conductance and soil–plant hydraulic conductance in pot-
grown maize (Cai et al., 2022b). Contrast results were found in winter wheat where plant hydraulic
conductance increased with rising VPD for some genotypes in wet conditions (Ranawana et al., 2021).
Vadez et al., (2021) examined the effects of soil types together with increasing VPD on transpiration
efficiency (TE) and yield under pot conditions for several $C_4$ species (maize, sorghum, and millet). The
interpretation of differences in TE was attributed to soil types, more specifically, to the differences in soil
hydraulic properties and soil hydraulic conductance However, experimental evidence linking root
hydraulics to stomatal regulation was lacking in these two Vadez's studies (Vadez et al., 2021).
Extrapolation and use of results obtained in pots or under greenhouse conditions to the field scale are
difficult due to the fact that soil substrates in pots might not represent natural soil in the field (Passioura,
2006). There is often greater evaporative demand and considerable fluctuation and interactions of climatic
variables in the field as compared to experiments under controlled or semi-controlled conditions. Recent
field studies have aimed at quantification of root hydraulic conductance and it linkages with crop growth
(leaf area and biomass) under different soil types (in wheat Cai et al., 2017; Cai et al., 2018; Nguyen et al.,
2020 or maize in Nguyen et al., 2022; Jorda et al., 2022). However, field studies that consider both below
(soil-root hydraulic conductance) and above (stem hydraulic conductance), or soil-plant hydraulic
conductance (includes below and above-ground parts) and their roles in stomatal regulation as well as
crop growth (leaf area and biomass) are rarely carried out.
This study aims at further understanding of the hydraulic linkages between soil and plant and responses
of plants to drought stress in relation to root: shoot growth characteristics at field scale. We hypothesize
that, in field-grown maize, (1) soil-plant hydraulic conductance depends on soil hydraulic properties,



especially under dry soil conditions (2) minimum leaf water potential of maize is similar across soil types,
water treatments and climatic conditions. The hypotheses will be tested through three objectives: (i) to
investigate the effects of soil types, water application, and climatic condition on root growth and (ii) on
stomatal conductance, leaf photosynthesis, transpiration, leaf water potential, different components
(root, stem and whole soil-plant hydraulic conductance), and (iii) to analyze the relative contribution of
root and shoot growth (leaf area and biomass) on the water uptake capacity of maize. These three
objectives will be achieved based on a comprehensive dataset covering the whole soil-plant continuum
over two growing maize seasons with contrasting climatic conditions (low and high VPD) under two water
treatments (rainfed and irrigated) and two different soil types (stony and silty soil).
**2. Materials and methods**
**2.1. Location and experimental set-up**
We carried out a field experiment at two rhizotron facilities in Selhausen, North Rhine-Westphalia,
Germany (50°52'N, 6°27'E). The field is slightly inclined with a maximum slope of around 4°. One rhizotrone
facility was located upslope (F1) with around 60% gravel by weight in the 10-cm topsoil while the second
rhizotrone facility was at downslope (F2) with silty soil (stone content is around 4% by weight).
Each experimental site was divided into three subplots of 7.25 m by 3.25 m: two rainfed plots (P1, P2), and
one irrigated plot (P3). In rainfed plots P1, other sowing densities and dates were used than in the other
plots and we excluded therefore these plots. Silage maize *cv.* Zoey was sown on 4 May and 8 May in 2017
and 2018, respectively, with a plant density of 10.66 seeds m$^{-2}$ (Figure 1a; Table 1). Detailed information
of crop management practices is provided in Table 1.
[Insert Table 1 here]



**2.2. Water applications**
Weather variables (global radiation, temperature, relative humidity, precipitation, and wind speed) were
recorded every 10 minutes by a nearby weather station (approx. 100 m from the experiment). Drip lines
(T-Tape 520-20-500, Wurzelwasser GbR, Müzenberg, Germany) were installed for irrigation at 0.3 m
intervals parallel to the crop rows. In 2017, maize received a total amount of 230 mm precipitation during
the growing period (136 days). Average, minimum and maximum daily air temperature were 17.6, 8.3, and
25.3 °C, respectively (Fig. 1b). The crop on P3 was irrigated (in total 130 mm) every 5-7 days (in total 10
times) using 13 mm of irrigation water per event between mid June to end of August for the irrigated plots
(2017F1P3 and 2017F2P3) (Fig. 1b). In 2018, average, minimum, and maximum daily air temperature were
19.2, 10.85, and 27.3 °C, respectively (Fig. 1b) and exceeded those of 2017. Characterized by exceptionally
hot and dry weather conditions, the summer season 2018 can be classified as an extreme year with respect
to plant growth at our site. Maize experienced high temperatures and VPD, especially around tasseling
and silking. In 2018, only 91.3 mm of rain were recorded in the growing period of 2018 (107 days). The
maize crop was irrigated every 5-7 days (in total 13 times), with a total amount of irrigation of 257 mm
and 239 mm between mid- June and mid- August for the irrigated plots 2018F1P3 and 2018F1P3,
respectively (Fig. 1d). In contrast to 2017, the rainfed plot in the stony soil (2018F1P2) had to be irrigated
(in total 66 mm) in four times (using 13, 22, 13, and 18 mm, respectively) to avoid a crop failure due to
severe drought (Fig. 1d).
[Insert Figure 1 here]
**2.3. Measurements**
**2.3.1. Soil water measurement and root growth**
At soil depths of 10, 20, 40, 60, 80, and 120 cm, MPS-2 matrix water potential and temperature sensors
(Decagon Devices Inc., UMS GmbH München, Germany) were installed to measure half-hourly soil water
potential and soil temperature. The range of the water potential measurements is form -9 kPa to



approximately -100000 kPa (pF 1.96 to pF 6.01). In addition to MPS-2, soil water potential was measured
by pressure transducer tensiometers (T4e, UMS GmbH, München, Germany) where the minimum
detectable suction is -85 kPa to +100 kPa. A detailed description of sensor installation, calibration and data
post processing can be found in Cai et al., (2016).
Minirhizotubes (7 m long clear acrylic glass tubes with outer and inner diameters of 6.4 and 5.6 cm,
respectively) were installed horizontally at six different depths of 10, 20, 40, 60, 80, and 120 cm below the
soil surface in each facility. There are three replicate tubes at each depth, accounting for 54 tubes in each
facility. Root measurements were taken manually by Bartz camera (Bartz Technology Corporation) (23
June 2017 – 12 September2017) and VSI camera (Vienna Scientific Instruments GmbH) (08 June 2017 – 22
June 2017) in 2017 while only VSI was used in 2018 (23 May2018 - 23 August 2018). Root images were
repeatedly taken from both left and right sides at 20 locations along horizontally installed minirhizotubes.
The root images were analyzed by automated minirhizotube image analysis pipeline for segmentation and
automated feature extraction (Bauer et al., 2021). Two-dimensional root length density (RLD, in units of
cm cm$^{-2}$) was estimated from the total root length observed in the image and the image surface area. The
overview of camera system, minirhizotube images acquisition, and post-processing of the root data were
described in detail in Bauer et al. (2021).
**2.3.2. Crop growth measurement**
The phenology, plant height, stem diameter, green and brown leaf area, dry matter of different organs,
and total aboveground dry matter were observed and measured bi-weekly. Plant height was measured of
15 randomly selected plants. The diameters of five randomly selected stems were measured. Due to the
limited number of plants in each plot, only two plants per measurement date were sampled to determine
total aboveground dry matter and leaf area (7 and 8 times in 2017 and 2018, respectively). Green and
brown leaf area was measured by a LI-3100C (Licor Biosciences, Lincoln, Nebraska, USA). At harvest, five





separate replicates (1m² each) were harvested. The dry matter of separate organs was determined after
drying at 105 °C for 48 hours (Nguyen et al., 2020).
**2.3.3. Leaf gas exchange, leaf water potential, and sap flow measurements**
Hourly leaf stomatal conductance (Gs), net photosynthesis (An), and leaf transpiration (E) were measured
every two weeks under clear sky conditions. Observations from 8 AM to 5 PM on four days and from 10
AM to 4 PM on six days were carried out in 2017. In 2018, measurements were carried out on 6 days from
8 AM to 7 PM and on 5 days from 10 AM to 4 PM (Nguyen et al., 2022a). The Gs, An, and E of two sunlit
leaves (uppermost fully developed leaves) and one shaded leaf of different plants were measured at
steady-state using a LICOR 6400 XT device (Licor Biosciences, Lincoln, Nebraska, USA). After leaf gas
exchange measurements, leaves were quickly detached using a sharp knife to measure leaf water potential
($\psi_{leaf}$)/ with a digital pressure chamber (SKPM 140/ (40-50-80), Skye Instrument Ltd, UK) with the working
air pressure ranging from 0 to 35 bars. To study the diurnal course of $\psi_{leaf}$ under dry and re-wetted soil
conditions, in 2018, measurements were undertaken for three additional days with predawn
measurements two days before and one day after irrigation. Further detail of measurement dates, range
of real time records of PAR, VPD and soil water status could be found in (Nguyen et al., 2022a).
In 2017 (from 7 July 2017 until harvest) and 2018 (from 28 June 2018 until harvest), 20 sap flow sensors
(SGA 13, SGB 16, and SGB 19 types) were installed (one sensor per plant and 5 maize plants per plot) based
on stem diameter size. Sensor data, in particular the partitioning of energy, electricity supply, sap flow,
and the temperature difference between upper and lower thermocouples (dT) of each sensor were
recorded at 10 minute intervals using a CR1000 data logger and two AM 16/32 multiplexers (Campbell
Scientific, Logan, Utah). The sap flow in the plant (g h⁻¹) was monitored directly by the data loggers
(Dynamax, 2007) and used as a surrogate for canopy transpiration based on the number of plants per
square meter.





**2.4. Calculation of total root length, root system conductance, stem, and whole plant hydraulic**
**conductance**
To estimate the total root length from minirhizotubes, we adopted the option 2 which was described in
Cai et al., (2017). Total root length per square meter soil surface area within each soil layer (m m$^{-2}$) was
computed by multiplying the root length density with the corresponding soil layer thickness. The root
length density was determined in each depth by dividing the measured root length per minirhizotron
image by the assumed volume the roots would have occupied in absence of the tube, i.e., W * L * tube
radius (see Cai et al., 2017).
Following Nguyen et al., (2020), the effective soil water potential was calculated based on hourly measured
soil water potential ($\psi_i$) and normalized root length density at six depths (10, 20, 40, 60, 80, and 120 cm)
(NRLD$_i$), and soil layer thickness ($\Delta z_i$) in the soil profile (Equation 1).

$$\psi_{soil\_effec} = \sum_{i=1}^{N} \psi_i NRLD_i \Delta z_i \tag{1}$$

We followed Ohm's law analogy by dividing the hourly sap flow by the difference between effective soil
water potential and shaded leaf water potential to estimate root system conductance (K$_{soil\_root}$ - Equation
2), between shaded leaf water potential and sunlit leaf water potential to estimate stem hydraulic
conductance (K$_{stem}$ - Equation 3), and between effective soil water potential and sunlit leaf water potential
to estimate whole plant hydraulic conductance (K$_{soil\_plant}$ - Equation 4).

$$K_{soil\_root} = Sapflow/(\psi_{soil\_effec} - \psi_{shadedleaf}) \tag{2}$$

$$K_{stem} = Sapflow/(\psi_{shadedleaf} - \psi_{sunlitleaf}) \tag{3}$$

$$K_{soil\_plant} = Sapflow/(\psi_{soil\_effec} - \psi_{sunlitleaf}) \tag{4}$$





During one measurement day, four values of the $K_{soil\_root}$, $K_{stem}$, and $K_{soil\_plant}$ were obtained from
measurements between 11AM and 2 PM. The average and standard deviation of these hourly
measurements were calculated for each measurement day in order to present the seasonal dynamics of
those variables. To capture the diurnal and seasonal variations of sap flow and sunlit leaf water potential,
in addition, we plotted the hourly sap flow and hourly difference of effective soil water potential and sunlit
leaf water potential for three measurement days starting from predawn and whole seasons, respectively,
to derive the slope which is also $K_{soil\_plant}$.
**2.5. Statistical analysis**

Regression analysis was performed to understand the relationship between the sap flow volume and the
difference of effective soil water potential and sunlit leaf water potential as well as the relationship
between the total aboveground biomass and cumulated water transpired (sap flow volume). These
analyses allow to derive the slope as proxy of $K_{soil\_plant}$ and transpiration use efficiency, respectively. Since
all measured data have their own measurement errors, the generalized Deming regression was employed.
We performed relationships (via correlation coefficient and statistical significant levels) of midday leaf An,
Gs, and E with midday $K_{stem}$, $K_{soil\_plant}$, $K_{soil\_root}$, sunlit leaf potential, $\psi_{soil\_effec}$, and the difference of $\psi_{soil\_effec}$
and sunlit leaf water potential ($\psi_{difference}$). All data processing and analysis were conducted using the R
statistical software (R Core Team, 2022).
**3. Results**
**3.1. Root growth under different water treatments, soil types and climatic conditions**
Observed root length (cm cm$^{-2}$) from the minirhizotubes in different soil depths at the first week of June
(stem elongation), around silking, and at harvest in two growing seasons are shown in the Figure 2. Root
length was similar among water treatments at the start of stem elongation in both years (Fig. 2a & 2d).
The difference in root length was pronounced at silking and harvest between the soil types. More root
growth was observed in the silty soil compared to the stony soil with the same water treatment (i.e. 2.5 -



6 times higher at depth 40 cm). This indicated the strong negative effects of stone content on root
development. In the stony soil, root length in the irrigated plot (F1P3) was slightly higher than in the rainfed
plot (F1P2). In contrast, the rainfed treatment (F2P2) in the silty soil showed much higher root length,
especially from 40 to 120 cm depths as compared to the irrigated plot (F2P3) in both growing seasons.
Much lower stone content and deep soil cracks in the silty soil (Morandage et al., 2021) allow root
extension to the subsoil, particularly in the rainfed plot F2P2. Root length in the rainfed treatment (F2P2)
in 2018, is higher than in 2017 which implies that root further developed to exploit the water in the soil
under the rainfed condition to meet the higher evaporative demand.
[Insert Figure 2 here]
Total root length (m m$^{-2}$) estimated from minirhizotubes and its ratio to shoot dry matter (m kg$^{-1}$) at three
measured dates (as in Figure 2) are shown in the Figure 3. Total root length was much higher for the silty
plots as compared to stony plots. In 2017, the highest total root length was observed in the rainfed plot of
the silty soil (F2P2) with approximately 9166 m m$^{-2}$ and 9878 m m$^{-2}$ around silking and harvest, respectively,
which was almost two times higher than in the irrigated plot (F2P3). These figures were higher in 2018
than 2017 where total root length of F2P2 was 10188 m m$^{-2}$ and 13750 m m$^{-2}$ at silking and harvest time,
respectively. For the rainfed stony soil (F1P2), soil water depletion around the beginning of June in 2017
(Supplementary material 1a) and from the first two weeks of June to harvest in 2018 (Supplementary
material 2a) caused the strong reduction of shoot biomass. In the stony soil, the shoot dry matter of the
irrigated plot (F1P3) and the rainfed plot (F1P2) were 1275 and 536 g m$^{-2}$ at silking time (e.g. 19 July 2018
–DOY 200, Supplementary material 3a and 3b). However, there was a minor difference between F1P2 and
F1P3 in terms of the ratio of root length to shoot dry matter. In the silty soil, a decrease of soil water
potential was not pronounced (compared to stony soil) in both years 2017 and 2018 (Supplementary
material 1b and 2b). In 2018, shoot biomass in the irrigated stony soil (F1P3) and silt soil (F2P3) were
similar (1275 and 1299 g m$^{-2}$, respectively on 19 July 2018 – DOY 200) while the shoot biomass of the



rainfed silty soil (F2P2) was 876 g m$^{-2}$ (Supplementary material 3a & 3b). However, the ratios of root length
to shoot biomass in the rainfed plot of the silty soil (F2P2) were 3 and 6 times higher than those in the
irrigated silty soil (F2P3) and stony soil (F1P3), respectively (e.g. 18 July, DOY 199). Moreover, total root
length was relatively equal among treatments at the start of set elongation (DOY 159, first week of June)
in both years, while this was the opposite for the ratio of root length to shoot dry matter. This firstly
illustrated that the finer soil texture without stones and with soil cracks could favor the root growth which
indicates strong interactions of root and soil conditions. Secondly, the larger root length and higher
atmospheric evaporative demand in 2018 than 2017 indicates also the interaction of root growth and
climatic conditions.
[Insert Figure 3 here]
**3.2. Stomatal conductance, photosynthesis, transpiration, and K$_{soil\_plant}$**
**3.2.1. Diurnal course of stomatal conductance, photosynthesis, transpiration, and water potential at leaf**
**level**
After a long period with high temperatures and no rainfall, soil water reduction in the rainfed plot of the
stony soil (F1P2) on 17 July 2018 (Supplementary material 2) resulted in three times lower net
photosynthesis (An), stomatal conductance (Gs), transpiration (E) and leaf water potential ($\psi_{leaf}$) as
compared to the remaining treatments (Fig. 4). This indicates that the soil water content strongly affected
the stomatal conductance. Stomatal closure was much pronounced around midday in F1P2 while this was
not the case in the F2P2, indicating the soil type strongly affected the stomatal conductance and leaf gas
exchange.
[Insert Figure 4 here]
Leaf gas exchange and leaf water potential in the F1P2 were still much lower than in other plots (Figure
5). On 18July 2018, after application of 22.75 mm of irrigation water (at 4 PM), photosynthesis, stomatal



conductance, transpiration and leaf water potential were slightly increased in F1P2. However, these were
still smaller than in F2P2 and the two irrigated plots.
[Insert Figure 5 here]
On the next day after irrigation, leaf gas exchange and water potential were considerably increased in the
F1P2 (Figure 6). Leaf curling was also less pronounced as compared the two previous days. This indicated
the recovery of plant after watering. Leaf water potential, photosynthesis, stomatal conductance, and leaf
transpiration were almost similar to other plots from predawn throughout the day.
[Insert Figure 6 here]
**3.2.2. Seasonal course of stomatal conductance, photosynthesis, transpiration, water potential, and**
**plant hydraulic conductance at the leaf level**
Seasonal stomatal conductance (Gs) and leaf water potential ($\psi_{leaf}$) are described in Figure 7. The
relationship between two variables was rather noisy and non-linear. The leaf water potential showed
distinct patterns among treatments in one growing season. Minimum $\psi_{leaf}$ was maintained at around -1.5
MPa in the irrigated plot in stony soil (F1P3) and two plots in the silty soil (F2P2 and F2P3). Lower minimum
$\psi_{leaf}$ could be observed in the rainfed plot with stony soil (F1P2) but it did not go beyond -2 MPa. Minor
leaf curling was observed only in the second week of June in the F1P2 in 2017. In 2018, the higher
temperature and vapor pressure deficit resulted in lower minimum $\psi_{leaf}$ in all treatments and soil types as
compared to 2017. The minimum $\psi_{leaf}$ was around -2 MPa in F1P3, F2P2, and F2P3 while $\psi_{leaf}$ could drop
below -2 MPa in F1P2 which was due to the severe soil water deficit. The low Gs and $\psi_{leaf}$ associated with
measurement dates when the substantial leaf curling was observed at mid of July to the end of growing
season in F1P2 in 2018 (Supplementary material 3c & 3d and Supplementary material 4c & d).
[Insert Figure 7 here]



The effective soil water potential ($\psi_{soil\_effect\ MD}$), sunlit leaf water potential ($\psi_{sunlitleaf\ MD}$), stomatal
conductance ($Gs_{MD}$), and whole plant hydraulic conductance ($K_{soil\_plant\ MD}$) at midday at several times during
the growing season are presented in Figures 8 and 9 for 2017 and 2018, respectively. As expected, there
was not much difference in terms of $\psi_{soil\_effecMD}$ among F1P3, F2P2, and F2P3 from 02 August to one week
before harvest in 2017. The lowest $\psi_{soil\_effec\ MD}$ was observed in the F1P2. Leaf water potential dropped
drastically but also $K_{soil\_plant\ MD}$ increased strongly whereas $\psi_{soil\_effec\ MD}$ remained quite similar (e.g. 18 July).
This is because sap flow have increased substantially in this day (e.g. from 2.34 mm d$^{-1}$ on 17 July to 6.97
mm d$^{-1}$ on 18 July for the F1P2). The stomatal conductance decreased a lot in this day which could be
explained that the atmospheric demand increased (e.g. global radiation was 13.6 MJ m$^{-2}$ on 17 July
compared to 23.9 MJ on 18 July while daily VPD was 0.7 kPa and 1.2 kPa, respectively) even more than the
sap flow. Midday sunlit leaf water potential was not distinctively different among treatments with the
lowest $\psi_{sunlitleaf\ MD}$ around -1.6 MPa throughout season. Also, $Gs_{MD}$ was rather similar among plots. The
$K_{soil\_plant\ MD}$ ranged from 0.125 to 0.96 mm h$^{-1}$ MPa$^{-1}$ with a sharp reduction before harvest. In general, the
lowest values of $K_{soil\_plant\ MD}$ were found in F1P2 which was consistent with the smaller overall seasonal
$K_{soil\_plant}$ (as the slope of linear relationship between sap flow and difference of effective soil water potential
and sunlit leaf water potential) (see Supplementary material 5).
[Insert Figure 8 here]
The $\psi_{soil\_effec\ MD}$ was substantially different in the two soil types and water treatments in 2018 (Figure 9a).
Both F1P2 and F1P3 showed a gradual drop of $\psi_{soil\_effec\ MD}$ from 15 June until the third week of July then
increased after irrigation events on 18 July (Supplementary material 2b). However, $\psi_{soil\_effec\ MD}$ of F1P2 was
much lower than F1P3 toward the harvest. The $\psi_{soil\_effec\ MD}$ of F2P2 and F2P3 only decreased progressively
from around 10 July till harvest even though there was water supply from the irrigation (Supplementary
material 2b). The water applied by irrigation and coming in by rainfall were insufficient to wet up the
deeper soil layers which remained dry. The low $Gs_{MD}$ was corresponding to the lowest $\psi_{sunlitleaf\ MD}$ and





$K_{soil\_plant\ MD}$ from the F1P2 (Figure 9c & 9d). The $K_{soil\_plant\ MD}$ from all plots was ranging from 0.12 to 0.91 mm
$h^{-1}$ MPa$^{-1}$.There was the drop in $K_{soil\_plant\ MD}$ (i.e. 3 to 9 July or 17-18 July) before irrigation in this plot.
However, it increased after the irrigation (i.e. 10 July and 19 July). This suggests that $K_{soil\_plant}$ depends
strongly on the soil water content and the conductivity of the rhizosphere.
[Insert Figure 9 here]
**3.2.3. Relationships of stomatal conductance, transpiration, photosynthesis with plant hydraulic**
**variables at the plant canopy level**
The slope of linear relationship between sap flow and difference of $\psi_{soil\_effec}$ and $\psi_{sunlitleaf}$ is shown for three
consecutive days (leaf water potential measurements from the predawn) and before and after irrigation
applications (17, 18, and 19 July 2018 or DOY 198, 199 and 200, respectively) (Figure 10). On both DOYs
198 and 199, the difference between $\psi_{soil\_effec}$ and $\psi_{sunlitleaf}$ was around -1.6 MPa with very low transpiration
rates in the treatment F1P2 which was associated with very low plant hydraulic conductance and leaf
curling. The whole plant hydraulic conductance was disrupted on these two days (0.06 and 0.16 mm $h^{-1}$
MPa$^{-1}$ for DOY 198 and 199, respectively). Water was supplied on DOY 199 at 1 PM for the irrigated plots
(F1P3, F2P3) as well as F1P2 at 4 PM (for saving plant from death due to severe drought stress). $K_{soil\_plant}$
was slightly changed (0.43 and 0.57 mm $h^{-1}$ MPa$^{-1}$ for F1P3 on DOY 199 and 200, respectively and 0.5 and
0.58 mm $h^{-1}$ MPa$^{-1}$ for F2P3 on DOY 199 and 200, respectively). However, the increase of $K_{soil\_plant}$ was
substantial in the F1P2 after the irrigation. Soil water replenishment and an increase in the root - soil
contact (Fig. 9a) allowed the $K_{soil\_plant}$ to recover overnight to 0.46 mm $h^{-1}$ MPa$^{-1}$. This resulted in a narrower
water potential gradient between root zone and sunlit leaf and in a higher transpiration rate on DOY 200.
[Insert Figure 10 here]
Seasonal average of different midday hydraulic conductance components (root system hydraulic
conductance - $K_{soil\_root}$, stem hydraulic conductance – $K_{stem}$, and whole plant hydraulic conductance –



$K_{soil\_plant}$) are shown in Figure 11. In the same year, the $K_{stem}$ was not much different among F1P3, F2P2, and
F2P3 plots. The $K_{stem}$ of those plots was slightly higher than in the F1P2 in both years. In general, the $K_{soil\_root}$
was lower than the $K_{stem}$. The $K_{soil\_root}$ in the F1P2 in 2018 was much lower than the remaining plots while
the $K_{soil\_root}$ was not much different among plots in 2017. Overall, the estimated $K_{soil\_plant}$ was around 1/
(1/$K_{soil\_root}$ +1/$K_{stem}$) regardless of soil types, years, and water treatments. Although there is a large
difference in total root length between two soil types (e.g. F1P3 versus F2P2), $K_{soil\_root}$ and $K_{soil\_plant}$ in those
two plots were not much different. The $K_{soil\_plant}$ and $K_{soil\_root}$ depend strongly on the soil water content and
the soil hydraulic properties. Therefore, $K_{soil\_plant}$ and $K_{soil\_root}$ were not only a plant property but also a soil
property.
[Insert Figure 11 here]
**3.3. Relative importance of root and leaf area growth to transpiration and crop performance at canopy**
**level**
Drought stress was observed in the rainfed plot (F2P2) in the second week of June 2017 with mild leaf
rolling. The crop then recovered due to sufficient rainfall and lower evaporative demand. Drought stress
occurring again at the stem elongation phase caused reduction of plant size (height and stem diameter)
(Supplementary material 4) as well as a slight reduction of leaf area and biomass in this plot
(Supplementary material 3a & 3c). Transpiration per unit of leaf area did not differ much among water
treatments and soil types in 2017 (Figure 12). The opposite was the case for the transpiration rate per unit
of root length. The observed root length at different soil depths (Figure 2) and total root length for two
plots in the stony soil was much smaller than in the silty soil (Figure 3). Therefore, transpiration per unit
of root length in the stony soils (F1P2 & F1P3) was almost 3 times higher than transpiration in the silty soil.
For the same soil, transpiration per unit root length of the irrigated treatment was slightly larger than in
the rainfed plot.



[Insert Figure 12 here]
The differences in sap flow per plant between water treatments and soil types were more pronounced in
2018 (Figure 13). The highest transpiration rate was observed in the irrigated plots (F1P3 & F2P3), followed
by the rainfed plot of the silty soil (F2P2) and it was lowest in the rainfed plot of the stony soil (F1P2).
These observations were in line with the differences in biomass and leaf area index between the
treatments (Supplementary material 3b & 3d) and plant size (Supplementary material 4b-c-d). In 2018,
severe leaf rolling was observed in the rainfed plot (F1P2) from the beginning of June until the end of the
growing period in 2018 (Supplementary material 3d). Similar to 2017, transpiration per unit of root length
was much higher in the stony plots as compared to silty plots. Also, for the silty soil, transpiration per unit
of root length of the irrigated plot (F2P3) was higher than in the rainfed plot (F2P2).
[Insert Figure 13 here]
Higher cumulative transpiration in the irrigated plots did not result in higher transpiration use efficiency
(TUE) in both soil types (Figure 14). For instance, TUE were 16.87 g mm$^{-1}$ and 15.59 g mm$^{-1}$ for F1P2 and
F2P2, respectively, while they were 15.47 and 14.79 g mm$^{-1}$ for F1P3 and F2P3, respectively, in 2017 (Figure
14A). For the same soil, the rainfed plot showed slightly higher TUE than the irrigated plot. When
comparing the TUE of maize of the two soil types for the same water treatment, TUE at the stony soil was
almost the same in silty soil. The TUE was not much different among treatments and soil types in 2018.
Overall, TUE in 2017 was higher as compared to 2018 (Fig. 14b).
[Insert Figure 14 here]
**4. Discussions**
**4.1. Effects of soil types, water application, and climatic condition on root growth**
Our root observations showed that soil type considerably affected root growth more than water treatment
(Figure 2). Root growth was strongly inhibited by the stony soil where much lower root length was



observed than in the silty soil, especially in the deeper soil layers. This was consistent with the findings
reported in (Morandage et al., 2021) where a linear increase of stone content resulted in a linear decrease
of rooting depth across all stone contents and developmental stages. Also, both simulations and
observations indicated that rooting depth was sensitive to the presence of cracks in the lower
minirhizontron facility (Morandage et al., 2021) which could explain the high root length between 40 and
120 cm soil depths which was observed in the silty soil in both years. In the silty soil, root growth was
favored towards deeper soil layers as also reported for the same field in 2016 for winter wheat (Nguyen
et al., 2020). Observation in field grown maize, the higher root length density and root diameter were
found in the sand than in the loam. This was attributed to the higher investment in nutrient exploration
because the lower concentration of plant-available nutrients was in sand than in loam (Vetterlein et al.,
2022). Also, the larger root diameters in sand than in loam are more likely explained by the need for soil
contact of the roots (Jorda et al., 2022; Vetterlein et al., 2022).
Our total root length was in the reported range of Cai et al., (2018) who studied winter wheat roots on the
same soil types in 2016. The total root length in our work was higher than the reported results from Cai
et al., (2018)  especially in the rainfed plot of the silty soil (F2P2) in 2018 (Fig. 3). In terms of the root: shoot
ratio, our observations were in line with those reported in the same soil types for wheat in Cai et al., (2018).
Ordóñez et al., (2020) has reported much larger figures of for instance 880 cm $g^{-1}$ in different locations and
under different N application rates in maize growing in the Midwest of US. Jorda et al., (2022) reported a
wide range of root: shoot ratio from 200 to 1000 cm $g^{-1}$ around flowering time of maize depending on the
wild type and root hair mutant genotypes growing on either loamy or sandy soils. More roots and higher
root: shoot ratios were found in the sand than in the loam in both wild type and root hair mutant
genotypes (Jorda et al., 2022; Vetterlein et al., 2022). Cai et al., (2018) observed much larger root: shoot
ratio in drought stressed plots than in irrigated plot in both soil types in winter wheat which indicated the
alternation of sink: source relationships to cope with water stress. This study emphasized that more



assimilates are used to promote root growth and extract more water under drought stress. However, this
was not the case for the stony soil in our work where the drought stress was more pronounced, especially
in 2018. A slightly higher root: shoot ratio in the F1P2 treatment compared to F1P3 (DOY 194 & 255) was
observed in 2017 while the root: shoot ratio in the two treatments was almost the same on DOY 199 and
228 in 2018 (Fig. 3). We only observed much higher root: shoot ratio in the rainfed plot (F2P2) as compared
to the irrigated plot on the silty soil (F2P3). Comas et al., (2013) has reported that maize increases the ratio
of root to leaf surface area and relative distribution to deeper depths in responses to water deficit. Under
drought stress, root growth of maize continues longer into the season than shoot and vegetative growth
and even beyond the onset of reproduction. A drop of soil water potential (Supplementary material 2b),
thus effective soil water potential (Figure 8a) was substantial from 10$^{th}$ July 2018 toward the harvest in the
rainfed plot in the silty soil (F2P2) which was consistent with the reduction of leaf water potential (Fig. 8b),
leaf area (Supplementary material 3c), total dry matter (Supplementary material 3d), and crop height
(Supplementary material 4b) as compared the irrigated plot (F2P3). This indicates a mild water stress in
2018 in the rainfed plots on the silty soil. The larger root: shoot ratio in this F2P2 plot in 2018 as compared
to F2P3 could be explained by the change of source: sink relations where more assimilates were devoted
to root growth, even at a later growth stage. Moreover, the low stone content and soil cracks (Morandage
et al., 2021) might favor root growth in the deeper soil layers which are close to the lower soil water table
in the site with silty soil (Vanderborght et al., 2010).
**4.2. Effects of soil types, water application, and climatic condition on stomatal conductance,**
**photosynthesis, transpiration, leaf water potential, and plant hydraulic conductance**
**4.2.1. Leaf water potential and stomatal conductance as affected soil water conditions**
In our study, stomata closed earlier and at more negative soil and leaf water potentials in stony soil than
in silty soil (see Fig. 4, 5, 6 and 7). In other work, Koehler et al., (2022) reported that maize stomata closed
at lower negative leaf water potentials in sand than in loam growing under controlled environment. Cai et





al., (2022b) investigated transpiration response of pot-grown maize in two contrasting soil textures (sand
and loam) and exposed to two consecutive VPD levels (1.8 and 2.8 kPa). Transpiration rate decreased at
less negative soil matric potential in sand than in loam at both VPD levels. In sand, high VPD generated a
steeper drop in stomatal conductance with decreasing leaf water potential which indicated the
transpiration and stomatal responses depend on soil hydraulics.
Stomatal control is an early and effective response to water stress to prevent the plant from water loss
and dehydration. Maize is considered as an isohydric plant which closes its stomata to maintain leaf water
potential above critical levels (Tardieu and Simonneau, 1998). Our results showed that minimum leaf
water potential varied among treatments (-1.5 MPa for F1P3, F2P2, and F2P3 and up to -2 MPa for F1P2
in 2017, while in 2018 minimum values were -2 MPa for F2P3, F2P2, and F2P3 and -2.7 MPa for F1P2) (Fig.
7 and Fig. 8, Fig. 9). Large variability of minimum LWP has been reported for maize genotypes. Leaf water
potential can be limited at quite high values, for instance -0.8 MPa in some lines of maize, while values as
low as -1.5 MPa have also been recorded (Welcker et al., 2011). Some drought-tolerant maize genotypes
close stomata at less negative leaf water potential under soil water depletion than more sensitive ones,
which is associated with their ability to avoid xylem embolism and hydraulic failure (Cochard, 2002; Tyree
et al., 1986; Li et al., 2009). However, our results show that the leaf water potential threshold can vary
within the same genotype depending on soil types, climatic conditions and water management. It should
be noted the constant $\psi_{leaf}$ level (around -1.8 MPa) under different soil water regimes reported in Tardieu
and Simonneau (1998) that was associated with high VPD values, was based on observations from a single
day. Measurements on $\psi_{leaf}$ and Gs for different days during several growing seasons have been rarely
reported for maize. The results of our study confirmed that maize appears to maintain its $\psi_{leaf}$ at around -
1.5 to -2 MPa which depended on evaporative demand and levels of soil moisture (Fig. 1, Fig. 7, Fig. 8, and
Fig. 9). This has been reported recently in Nguyen et al. (2022a). Our current study, which investigates the
drivers of the modifications of $\psi_{leaf}$ during the growing season, also confirmed that such stomatal



regulation and the $\psi_{leaf}$ were mediated by soil hydraulics. Cochard, (2002) reported that stomatal closure
is complete between -1.6 and -2 MPa. In our study, the observed $\psi_{leaf}$ was below -2 MPa for several days.
Similar values were also reported by Li et al. (2002) for field-grown maize in semiarid conditions. In our
study, leaf water potential dropped below -2 MPa in the rainfed plots to levels much lower than those
observed in the irrigated plots in 2018. This could imply different degrees of isohydry in maize. A
continuum exists in the degree to which stomata regulate the $\psi_{leaf}$ for trees (Domec and Johnson, 2012;
Klein, 2014) or in grape-vine (Schultz, 2003). Also, cultivars of grape vine show large differences in
minimum $\psi_{leaf}$ indicating differing degrees of isohydric behavior (Coupel-Ledru et al., 2014). When
comparing different herbaceous species, Turner et al., (1984) showed that there was a range of isohydric
behavior among the species in terms of the response to increasing vapor pressure deficit (VPD) under
sufficient soil moisture. However, conclusions concerning contrasting minimum $\psi_{leaf}$ between 2017 and
2018 should not be overemphasized. Observed extremely low $\psi_{leaf}$ correspond with the extremely low Gs
and were further accompanied by complete leaf curling in rainfed treatment under stony soil in 2018 (Fig.
4, 5, and Fig. 9) due to the extremely dry and hot summer and severe soil dryness.
**4.2.2. Hydraulic conductance components as affected by soil water conditions**
Estimates of hydraulic components in soil-plant-atmosphere continuum are important not only to
understand its underlying relationship to other crop characteristics (stomatal conduction, transpiration,
and photosynthesis) but also to provide modeling parameters in process-based soil-root-shoot models
(Nguyen et al., 2020; Sulis et al., 2019; Nguyen et al., 2022b). Measurement of the components of hydraulic
conductance are challenging under field conditions because it requires the estimate of transpiration and
root to leaf water potential gradients. To our knowledge, our results were unique with regard to the
dynamics of $K_{soil\_plant}$ for field-grown maize on two soil types and under contrasting water, and climate
conditions. Our seasonal $K_{soil\_plant}$ ranged from 0.12 mm h$^{-1}$ MPa$^{-1}$ to 0.9 mm h$^{-1}$ MPa$^{-1}$ (Fig. 8 & Fig. 9; Fig.
10, and Supplementary material 5). Root system hydraulic conductance ranged from 0.26 to 1.47 mm h$^{-1}$



MPa $^{-1}$ (Figure 11). Note that the unit of $K_{soil\_plant}$ as mm h$^{-1}$ MPa $^{-1}$ could be equivalent to the unit of 10$^{-5}$ h$^-$
$^1$ if one assumes 1MPa is approximately 10$^5$ mm in terms of pressure head. Cai et al., (2018) reported root
hydraulic conductance in winter wheat from 0.05 x 10$^{-5}$ h$^{-1}$ to 0.5 x 10$^{-5}$ h$^{-1}$ in two similar soil types. Nguyen
et al., (2020) also reported $K_{soil\_plant}$ in winter wheat from 0.0625 x 10$^{-5}$ h$^{-1}$ to 0.461 x 10$^{-5}$ h$^{-1}$. Meunier et al.,
(2018) focused on estimating the root system hydraulic conductance of maize in a container experiment
where the range of $K_{soil\_plant}$ was much larger from 0.37 x 10$^{-5}$ h$^{-1}$ to 36 x 10$^{-5}$ h$^{-1}$ for the plant density of 10
plant m$^{-2}$. Jorda et al., (2022) estimated root system hydraulic conductance of 0.5 to 1.5 10$^{-3}$ d$^{-1}$which
would be roughly between 2 to 6 10$^{-5}$ h$^{-1}$. To simulate leaf water potential in the modeling work for field
maize, Nguyen et al., (2022b) based on assumption that $K_{soil\_plant}$ was 0.53 x $K_{soil\_root}$. Such fraction (0.53) was
consistent with the reported range in our work (0.3-0.8) with average $K_{soil\_plant}$ was at half of root system
hydraulic conductance across treatments. In our work, except the F2P2 in 2018, the stem hydraulic
conductance was 10% to 60% higher than root system hydraulic conductance. Gallardo et al., (1996)
reported that stem hydraulic conductance of wheat was lower than root system conductance at around
71 to 91 days after sowing (DAS), but they were similar at 102 DAS. In lupine, stem hydraulic conductance
was two times higher than root system conductance regardless of measured days. The larger root length
in wheat than lupine did not necessarily result in higher root conductance in wheat. Together with this
study, our study emphasizes the values of stem hydraulic conductance compared to the root hydraulic
conductance in maintaining water potential gradient from shaded leaf or plant color to the sunlit leaf.
Our results showed clear differences in $K_{soil\_plant}$ among treatments where much lower $K_{soil\_plant}$ was
observed in the F1P2 as compared to F2P2 (see Figure 10 for 2018; Figure 8 and 9 and Supplementary
material 5 for both years). This indicated the soil texture dependence for whole plant hydraulic
conductance. Maize plants with the shorter root system (i.e. rainfed plot in the stony soil in 2018) (Fig. 3)
had lower plant hydraulic conductance. Our results indicated that there was an impact of soil hydraulic
conditions on $K_{soil\_plant}$ via the reduction of root system hydraulic conductance. Our analysis for three



consecutive measurement days in 2018 (Fig 10) showed that in the silty soil, $K_{soil\_plant}$ decrease when soil
water potentials are becoming more negative. For instance, in the silty soil in 2018 when the soil water
potentials were considerably lower in the rainfed than in the irrigated plot (e.g. after 10$^{th}$ July), $K_{soil\_plant}$
was lower in the rainfed than in the irrigated plot. In the stony soil, the $K_{soil\_plant}$ and leaf water potentials
seems to decrease more considerably (compared to the silty soil) when the soil water potentials become
more negative. In other words, $K_{soil\_plant}$ increased considerably when the soil water potentials in the stony
soil increased. Koehler et al., (2022) analyzed the maize plant responses to soil drying under controlled
climate conditions with three soil types (sand, sandy loam, and loam). This study confirmed the impact of
soil texture on plant response to soil drying in various relationships. In their work, the soil-plant
conductance decreased in both sand and loam but at less negative water potentials in the sand than in the
loam. Root system hydraulic conductance decreased at less negative bulk soil water potential in the coarse
soil than in the fine soil (Vanderborght et al., 2023).In our work, $K_{soil\_plant}$ increased slowly after irrigation
mainly for the severe water stress plot (see F1P2 on 19 July  in Fig 9d and 10c). This implied that added soil
water by irrigation took some time for recovery the soil-root contact within the rhizosphere.
**4.2.3. Relationships of stomatal conductance, transpiration, photosynthesis with plant hydraulic**
**variables**
In 2017, our estimated midday effective soil water potential ($\psi_{soil\_effec\ MD}$) did not vary much (between soil
types and treatments) which was consistent with the low variability in midday sunlit leaf water potential
($\psi_{sunlitleaf\ MD}$) and $K_{soil\_plant}$ among water treatments (Fig. 8). The $\psi_{soil\_effecMD}$ was high (around -0.35 MPa)
while $\psi_{sunlitleaf\ MD}$ was around -1.5 MPa (Fig. 8c). In contrast, the difference of $\psi_{soil\_effec\ MD}$, $\psi_{sunlitleaf\ MD}$, and
$K_{soil\_plant}$ was higher among water treatments and soil types in 2018 as compared to 2017. Moreover, the
high VPD and air temperature in combination with the small precipitation  in the main growing season in
2018 led to a stronger reduction of $\psi_{soil\_effec\ MD}$ up to -0.75 MPa (i.e. in F1P2 in the stony soil on 17 and 18
July in 2018, Figure 9) and $\psi_{sunlitleaf\ MD}$ to -2.5 MPa. This low $\psi_{soil\_effec\ MD}$ in F1P2 was associated with low



stomatal conductance (Fig. 9c), low $K_{soil\_plant}$ (Fig. 9d), and strong transpiration reduction (Fig. 10a-b, Fig.
12, and Supplementary material 5). Our results were in line with the analysis from Cai et al., (2022a) which
revealed that water uptake depended on effective soil water potential which in turn depended on soil
water potential which differed between plots with different textures.
The transpiration rate and $K_{soil\_plant}$ (slope of linear regression lines in Fig. 10a and b) were very low in the
rainfed plot under the stony soil (F1P2) which was associated with the large $\psi_{difference}$ (Fig. 10a & b) and the
lower stomatal conductance as compared to other plots (Fig. 9c). The $K_{soil\_plant}$ slightly increased after
irrigation (DOY 199 in Fig. 10b) corresponding with the smaller $\psi_{difference}$ (Fig. 10b) and an increase in
stomatal conductance (Fig. 9c). Seasonal $K_{soil\_plant}$ was low in the rainfed plot under stony soil (F1P2) with
the larger $\psi_{difference}$ (Supplementary material 5). In addition, our study showed that the midday stomatal
conductance, photosynthesis, and transpiration were significantly correlated only with midday $K_{soil\_plant}$ in
the rainfed plot on the stony soil (F1P2) in 2018 where high VPD and temperature occurred
(Supplementary material 6, 7, and Supplementary material 8). Maize plant had lower plant hydraulic
conductance in the rainfed plot in stony soil required the larger gradients in soil water pressure to sustain
the same transpiration rate (thus exhibited earlier stomatal closure) as compared to the same plot in the
silty soil. This was in line with a study from Abdalla et al., (2022) which suggested that during soil drying,
stomatal regulation of tomato is controlled by root and soil hydraulic conductance. Recent work from
Müllers et al., (2022) on faba bean and maize suggested that differences in the stomatal sensitivity among
plant species can be partly explained by the sensitivity of soil-plant hydraulic conductance to soil drying.
The loss of conductance has immediate consequences for leaf water potential and the associated stomatal
regulation. Cai et al., (2022b) also showed that the decrease in sunlit leaf stomatal conductance was well
correlated with the drop in soil-plant hydraulic conductance, which was significantly affected by soil
texture. This was confirmed in our work where the stony soil strongly impacted on root growth, modulated
$K_{soil\_plant}$, and consequently influenced the leaf stomatal conductance, photosynthesis, and transpiration.



**4.3. Relative contribution of water control by leaves and roots on transpiration and transpiration use efficiency**

Responses of crops via stomatal control to reduce water loss at leaf scale while maintaining leaf photosynthesis and water use efficiency were reported earlier (Nguyen et al., 2022a; Vitale et al., 2007). In addition to that, in the maize experiments in 2017 and 2018 leaf rolling was observed in both rainfed plots on the stony and the silty soil in the second week of June 2017 and from the beginning of June until the end of the growing period in 2018. This indicates another dehydration avoidance mechanism resulting from morphological adjustments which is an effective mechanism for delaying senescence (Aparicio-Tejo and Boyer, 1983; Richards et al., 2002). Stomatal closure resulted in more reduction of transpiration and assimilation in the rainfed plots than irrigated plots with the same soil type (Fig. 5, Fig. 6, Fig. 7, and Fig. 13A). There was reduction of shoot biomass (also stem size and leaf size adjustments) in F1P2 as compared to other plots. However, the TUE was not smaller in this plot than the remaining plots. These observations confirm that plant size adjustments through reduction of height, leaf width and length are efficient responses to reduce water loss at canopy scale in addition to stomatal control at the leaf level.

Relative contribution of leaf area to transpiration has been highlighted in wheat where reduction of tiller number resulted in significantly (lower LAI, thus lower canopy transpiration (Cai et al., 2018; Trillo and Fernández, 2005; Nguyen et al., 2022a). However, root system conductance per unit of leaf area and per unit root mass were strongly reduced and eventually more than reduction of leaf area under water stress (Trillo and Fernández, 2005). In our work, expressing the transpiration per unit of root length on the one hand allowed to analyze the role of total root length to water uptake. However, on the other hand, the lower total root length did not necessarily result in a lower root water uptake and vice versa. For instance, the rainfed plot of the treatment F2P2 had the larger total root length which could postpone the effect of soil water limitations in drying soils due to greater ability to extract water from subsoils. Therefore, transpiration was very similar between F2P2 and F2P3. Despite of the much lower total root length in the





stony soil, $K_{soil\_plant}$ in the irrigated plot (F1P3) was not much lower than in the same water treatment in
the silty soil (F2P3, Fig. 8c, 9c, Fig. 10, and Supplementary material 5). This could be explained by the fact
that the $K_{soil\_plant}$ variability was not only depended on root architecture (here the root length and
distribution) but also depended on the variability of root segment hydraulic properties which has also been
illustrated and discussed in Zwieniecki et al. (2002), Frensch and Steudle (1989), Meunier et al. (2018),
Couvreur et al. (2014), and Ahmed et al. (2018). Meunier et al. (2020) showed that more than 65% of the
variability of root system conductance of maize plants could be attributed to variability in root
architecture, which includes root length, whereas only 25% of the variability was attributed to root
segment hydraulic properties. However, the analysis of Meunier et al., (2020) neither included the impact
of root hairs nor the impact of rhizosphere conductivity but only focused on the root system hydraulic
conductance. Moreover, the contribution of shoot hydraulic conductance could be large in plants (Gallardo
et al., 1996; Trillo and Fernández, 2005; Sunita et al., 2014) which also confirmed in our work. In our work,
$K_{soil\_plant}$ comprised root and shoot conductance which are directly influenced by soil hydraulics. Our
estimates of $K_{soil\_plant}$ varied with transpiration and gradients of $\psi_{sunlitleaf}$ and $\psi_{soil\_effec}$. Thus, any change of
soil hydraulic conductance will change the root to shoot water potential. Consequently, it will affect the
gradients between shoot and root rhizosphere (Carminati and Javaux, 2020). Thus, our study is revealing
the importance of both soil texture characteristics and root phenotypic traits (here root length) in
regulating plant transpiration (Cai et al., 2022a). Other traits like root hair density ( Cai et al., 2022a) or
higher root length density (Vadez, 2014) could contribute to the soil to root water potential and root-zone
hydraulic conductance where dense root hairs are delaying soil water deficit in drying soils. However,
contrast results have shown that root hairs did not have an effects on root water uptake (see Jorda et al.
2022). The role of root hairs could not be analyzed in our work which based on the root data from
minirhizotron images.
**5. Conclusion**



We presented plant hydraulic characteristics and crop growth from root to shoot of maize under field-
grown conditions with two soil types (silty and stony), each soil with two water regimes (irrigated and
rainfed) for two growing seasons (2017, 2018). Our results confirmed that root length and root: shoot ratio
was modulated by soil types and water treatment but less by seasonal evaporative demand. Increase root:
shoot ratio has been an important response of maize that allows plants to extract more water under
drought stress that occurred rather in the silty soil but less in the stony soil due to the higher content of
stony material. Despite of lower root length in the stony irrigated plot, transpiration rate was not much
lower than in the silty irrigated plot. This could be related to another property of the root such as root
segment conductance or other root traits (e.g. root hair). Further investigation with extensive
measurements of roots including axial and radial root conductance at field scale will be required to better
explain the observed results.
Another conclusion is that stomatal regulation maintains leaf water potential at certain thresholds which
depends on soil types, soil water availability, and seasonal atmospheric demand. The stomata conductance
was smaller and at more negative leaf water potentials in stony soil than in silty soil. The leaf water
potentials are affected by the soil-plant plant hydraulic conductance. In addition to stomatal regulation,
leaf growth and plant size adjustments are important to regulate the transpiration that water use
efficiency was not different among treatments and soil types in the same year.
The lowest soil-plant hydraulic conductance was observed in the stony soil with severe drought stress as
compared to silty soil while its variation depends also on the soil water variation (before and after
irrigation). Root system and soil-plant hydraulic conductance depended strongly on soil hydraulic
properties. In the stony soil, which has a considerably smaller water holding capacity than the silty soil,
root length was considerably smaller than in the silty soil. Nevertheless water uptake per unit root length
was much larger than in the fine soil. This also means that the hydraulic conductance per unit root length
must have been much larger in the stony soil than in the fine soil. Cai et al., (2018) observed a similar effect



for winter wheat but they found much smaller differences in the root length normalized root conductance.
The higher root length normalized root conductance means that the anatomy of the root tissues must
have been influenced by the soil texture and compensated the considerably smaller root length in the
stony soil. Looking at the effect of water treatments in the silt soil, the non-irrigated plot had more roots
than the irrigated one and both had more roots in the year with high VPD. But the soil-root conductance
was higher in the irrigated plot than in the rainfed plot. This means that in the irrigated plot, the soil-root
conductance per unit root length was higher than in the rainfed plot. This could either be due to wetter
soil conditions and higher soil conductance or it could be due to a larger conductance of the root tissues.
Especially in 2017 when the silty soil was wetter, the slightly larger soil-root conductance in the irrigated
plot is most likely the result of larger root tissue conductance in the irrigated plot. Thus, how root
architecture (here represented simply by the total root length) and root tissue conductivities 'respond' to
drought stress might be opposite depending on the comparisons that are made. When the stony soil and
silt soil are compared, the higher 'stress' due to lower water availability in the stony soil resulted in less
roots with a higher root tissue conductance in the soil with more stress. When comparing the rainfed with
the irrigated plot in the silty soil, the higher stress in the rainfed soil resulted in more roots with a lower
root tissue conductance in the treatment with more stress. This illustrates that the 'response' to stress can
be completely opposite depending on conditions or treatments that lead to the differences in stress that
are compared. Therefore, it cannot be the 'stress' alone that defines how a plant will react and adapt its
root system. Modelling the impact of stress and the feedback between drought stress and plant
development is likely controlled by other properties or parameters that change with changing soil water
availability and atmospheric water demand then the plant stress level. Results from this study show that
soil-crop models should focus not only on simulating stomatal regulations to capture the response to
drought stress, but also require adequate representations of leaf growth and adjustments.



**List of Tables**

Table 1. Crop phenology and management information for different treatments in 2017 and 2018.

| | 2017 | | | | 2018 | | | |
|---|---|---|---|---|---|---|---|---|
| Soil types | Stony (F1) | Stony (F1) | Silty (F2) | Silty (F2) | Stony (F1) | Stony (F1) | Silty (F2) | Silty (F2) |
| Water treatments | Rainfed (P2) | Irrigated (P3) | Rainfed (P2) | Irrigated (P3) | Rainfed (P2) | Irrigated (P3) | Rainfed (P2) | Irrigated (P3) |
| Plot names | F1P2 | F1P3 | F2P2 | F2P3 | F1P2 | F1P3 | F2P2 | F2P3 |
| Growing season (days)$^{¥}$ | 136 | 136 | 136 | 136 | 107 | 107 | 107 | 107 |
| Cumulative rainfall (mm)$^{*}$ | 248.7 | 248.7 | 248.7 | 248.7 | 91.3 | 91.3 | 91.3 | 91.3 |
| Irrigation (mm) | 0 | 130 | 0 | 130 | 66 | 257.6 | 0 | 257.6 |
| Fertilizer application (mm/dd) (per hectare) | 05/09:100 kg N + 40kg $P_2O_5$ 07/06: 80 kg N + 40 kg $K_2O$ | | | | 05/22: 100 kg N 05/30: 40 kg $P_2O_5$ + 40 kg $K_2O$ 06/27: 80 kg N | | | |
| Sowing date (mm/dd) | 05/04 | | | | 05/08 | | | |
| Emergence date | 05/09 | | | | 05/13 | | | |
| Tasseling date | 07/09 | | | | 07/09 | | | |
| Silking date | 07/14 | | | | 07/11 | | | |
| Harvest date | 09/12 | | | | 08/22 | | | |

Notes: $^{¥}$ from sowing to harvest; $^{*}$ for rainfall for whole growing season;





**List of Figures**

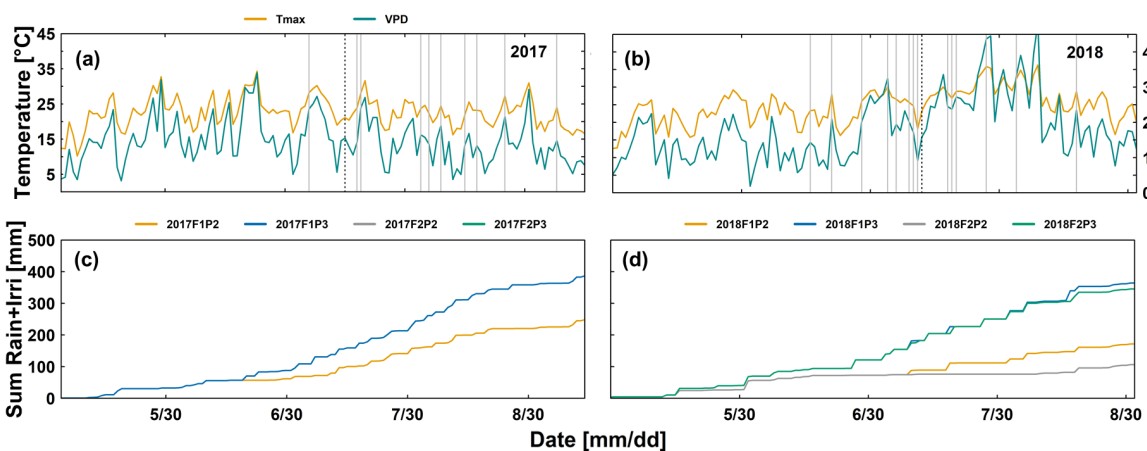

Figure 1: Daily maximum air temperature (Tmax) (°C), daily maximum air vapor pressure deficit (VPD) (kPa) in the two growing seasons (a) 2017 and (b) 2018 and cumulative (sum) of rainfall and irrigation from the rainfed (P2) and irrigated (P3) plots of the stony soil (F1) and silty soil (F2) in the two growing seasons (c) 2017 and (d) 2018. The black dashed vertical lines (a) and (b) indicate silking time. Grey vertical lines in (a) and (b) indicate the measured days for leaf gas exchange and leaf water potential. Two lines for 2017F2P2 and 2017F2P3 were overlapped by the lines from 2017F1P2 and 2017F1P3, respectively





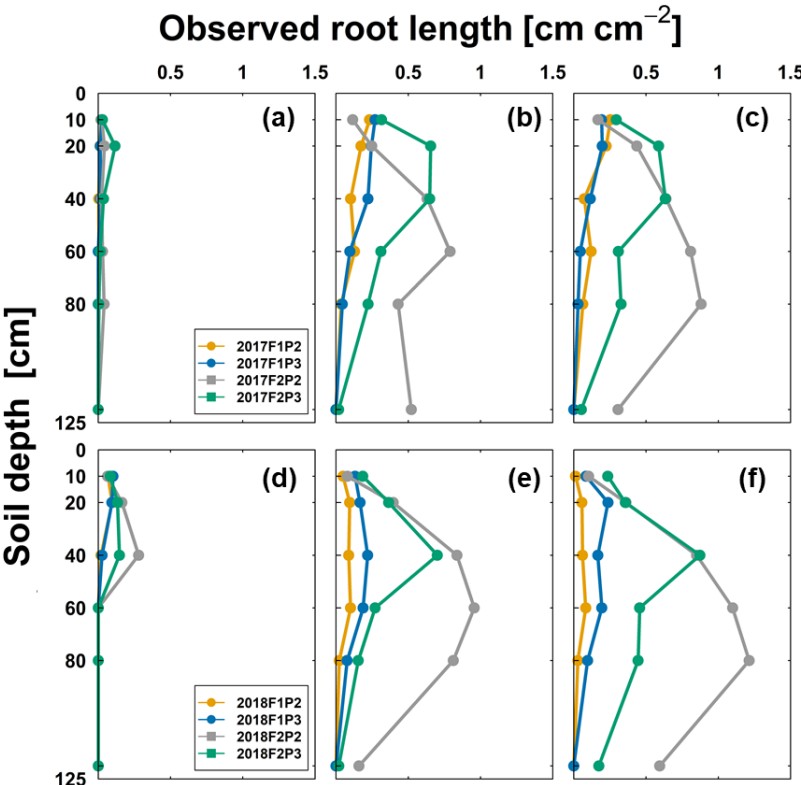

Figure 2: Observed root length from minirhizotubes (cm cm$^{-2}$) from 10, 20, 40, 60, 80, and 120 cm soil depth from the rainfed (P2) and irrigated (P3) plots of the stony soil (F1) and silty soil (F2) in the two growing seasons in 2017 (a - 8 June, b - at silking on 13 July, c - at harvest on 12 September) and in 2018 (d - 7 June, e - at one week after silking - 18 July, f - one week before harvest - 16 August).



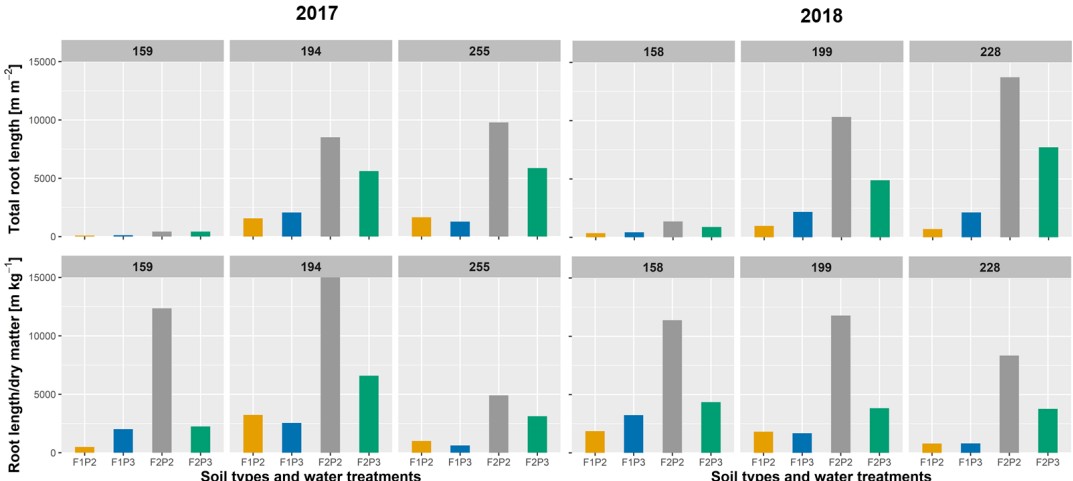

Figure 3: Observed root length from minirhizotubes (m m⁻²) and ratio of root length per shoot dry matter (m kg⁻¹) from the rainfed (P2) and irrigated (P3) plots of the stony soil (F1) and silty soil (F2) in the two growing seasons (DOY 159, 194, and 255, left panel) in 2017 and in 2018 (DOY 158, 199, and 228, right panel) where on 8 June (DOY 159) at silking on 13 July (DOY194) 2017; and at harvest on 12 September (DOY 255) in 2017; 7 June (DOY 158), one week after silking on 18 July (DOY 199); and one week before harvest on 16 August (DOY 228) in 2018 (see also Figure 2).



Figure 4. Diurnal course of (a) photosynthetically active radiation (PAR) and vapor pressure deficit (VPD), (b –e) leaf net photosynthesis (An), (f –i) leaf stomatal conductance (Gs), (j –m) leaf transpiration (E), and (n –q) leaf water potential (LWP) on 17 July in maize in 2018 before irrigation at the rainfed (P2) and irrigated (P3) plots of the stony soil (F1) and silty soil (F2). Measurement was carried out from shaded leaf (plus symbol with lines) and two sunlit leaves (solid dot - lines and solid square - lines).





Figure 5. Diurnal course of (a) photosynthetically active radiation (PAR) and vapor pressure deficit (VPD), (b –e) leaf net photosynthesis (An), (f –i) leaf stomatal conductance (Gs), (j –m) leaf transpiration (E), and (n –q) leaf water potential (LWP) on 18 July in maize in 2018 before irrigation at the rainfed (P2) and irrigated (P3) plots of the stony soil (F1) and silty soil (F2). Measurement was carried out from shaded leaf (plus symbol with line) and two sunlit leaves (solid dot - lines and solid square - lines). Crop was irrigated at 1 PM, 1 PM, 4 PM for F1P3, F2P3, and F1P2, respectively (22.75 mm for each plot) (Supp. 2).





Figure 6. Diurnal course of (a) photosynthetically active radiation (PAR) and vapor pressure deficit (VPD), (b –e) leaf net photosynthesis (An), (f –i) leaf stomatal conductance (Gs), (j –m) leaf transpiration (E), and (n –q) leaf water potential (LWP) on 19 July in maize in 2018 after irrigation at the rainfed (P2) and irrigated (P3) plots of the stony soil (F1) and silty soil (F2). Measurement was carried out from shaded leaf (plus symbol with line) and two sunlit leaves (solid dot - lines and solid square -lines). Crop was irrigated on 18 July at 1 PM, 1 PM, 4 PM for F1P3, F2P3, and F1P2, respectively (22.75 mm for each plot) (Supp. 2).



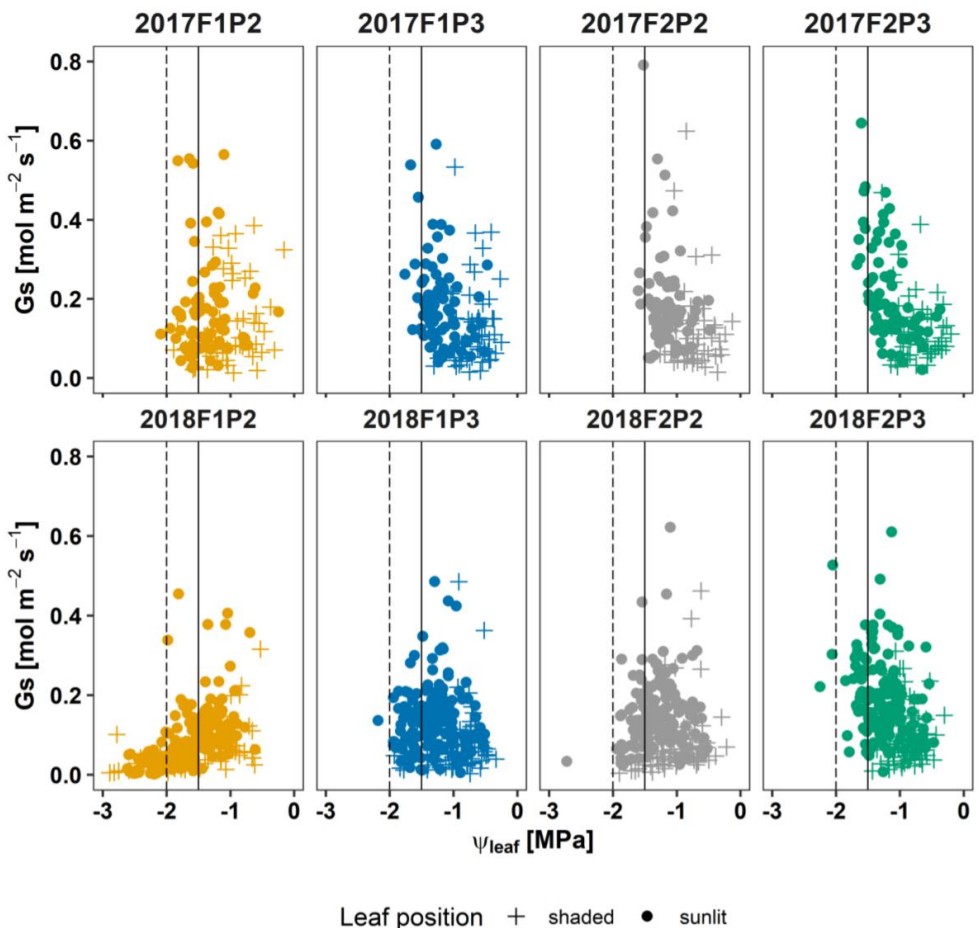

Figure 7: Seasonal stomatal conductance to water vapor (Gs) versus leaf water potential ($\psi_{leaf}$) in 2017 (top panel) and in 2018 (bottom panel) at the rainfed (P2) and irrigated (P3) plots of the stony soil (F1) and silty soil (F2). Vertically continuous and dashed lines indicated $\psi_{leaf}$ at -1.5 and -2 MPa, respectively. Measurement was carried out from shaded leaf (plus symbol) and two sunlit leaves (solid dots)



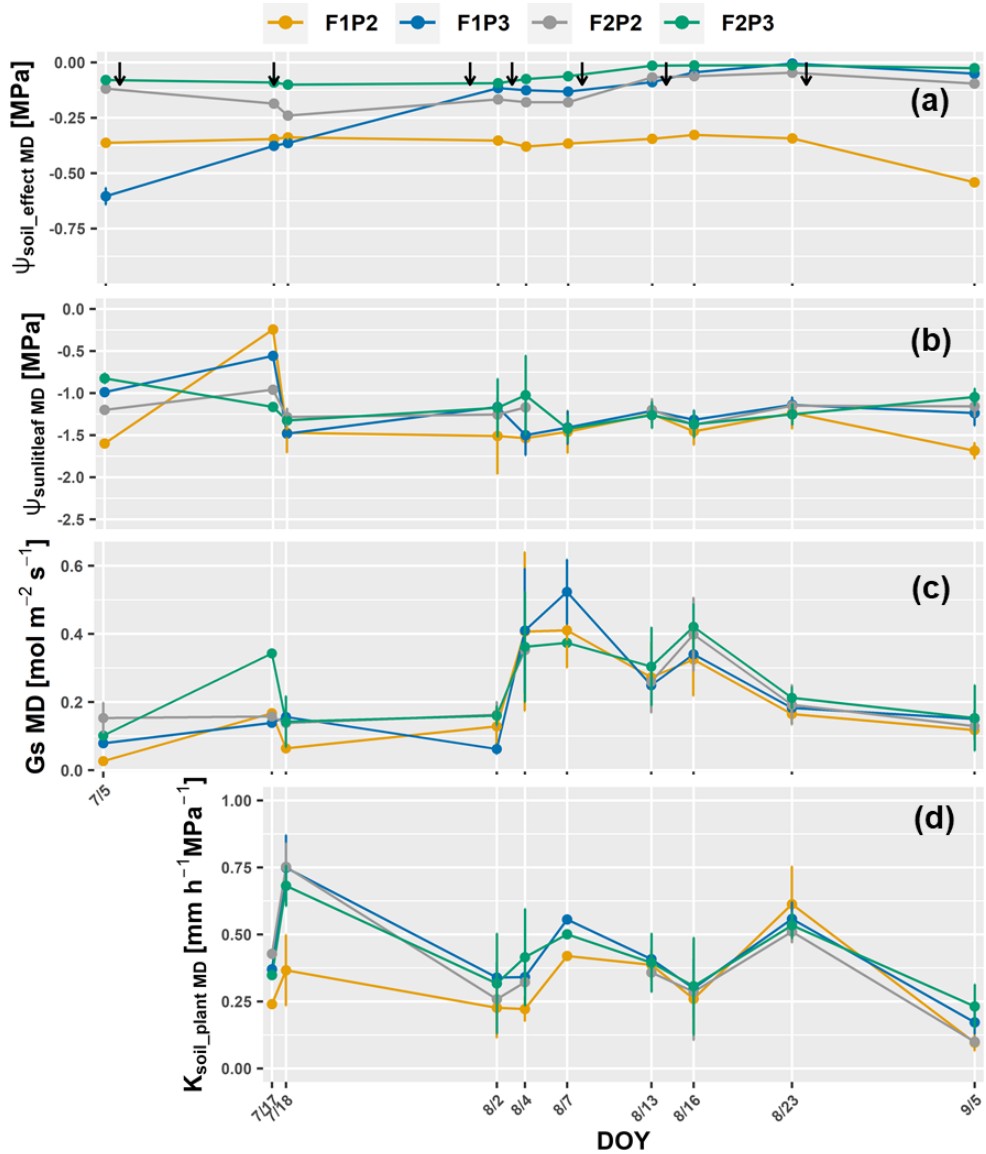

Figure 8: Dynamic of around midday (MD) of (a) the effective soil water potential ($\psi_{soil\_effec, MD}$) (b) sunlit leaf water potential ($\psi_{sunlitleaf MD}$), (c) stomatal conductance (Gs MD) and (d) whole soil-plant hydraulic conductance ($K_{soil\_plant MD}$) in the growing season 2017 from the rainfed (P2) and irrigated (P3) plots of the stony soil (F1) and silty soil (F2). Error bars indicate the standard deviation of the different values taken around midday (11 AM, 12AM, 1PM, and 2 PM) of different sunlit leaves. Whole soil-plant hydraulic conductance was shown from 17 July when sap flow was measured. The black arrows indicates the irrigation events for the irrigated treatments F1P3 and F2P3 in the showing period.



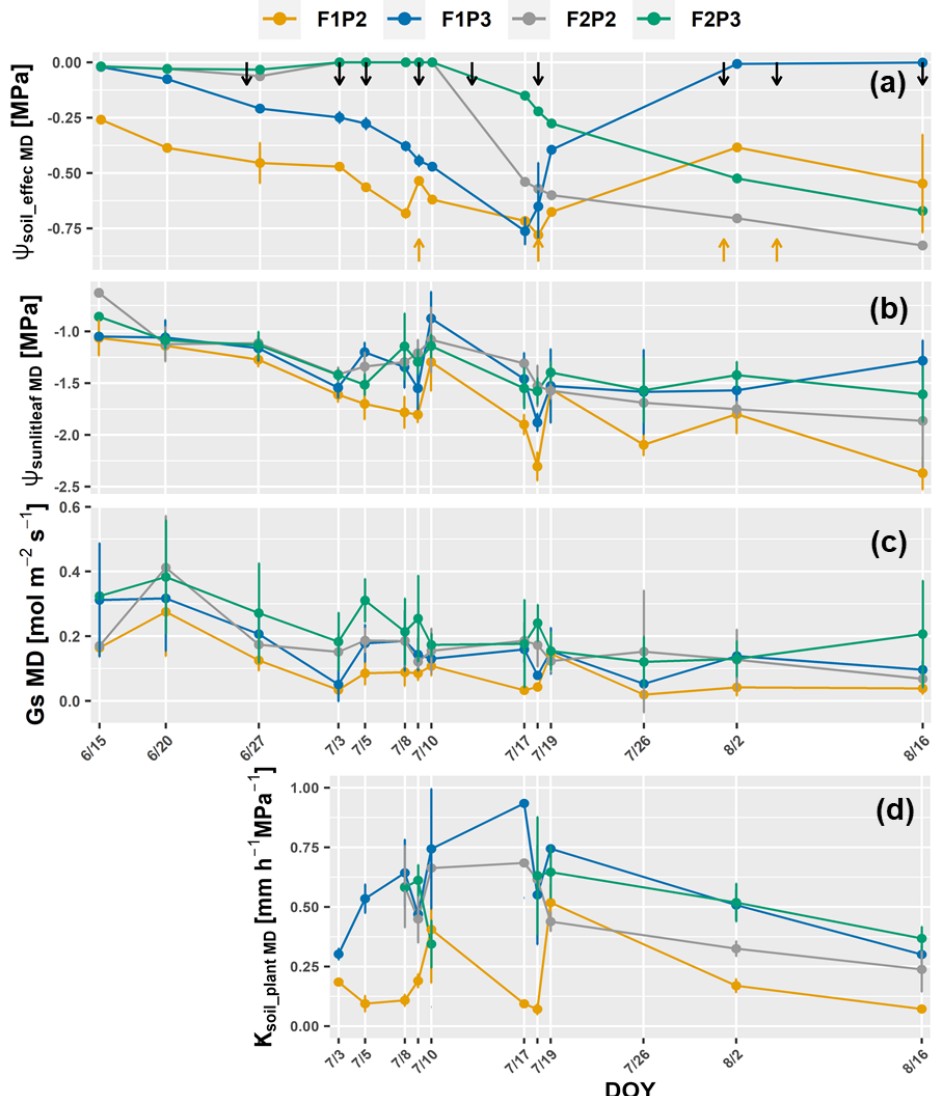

Figure 9: Dynamic of around midday (MD) of (a) the effective soil water potential ($\psi_{\text{soil\_effec MD}}$) (b) sunlit leaf water potential ($\psi_{\text{sunlitleaf MD}}$), (c) stomatal conductance (Gs MD) and (d) whole soil-plant hydraulic conductance ($K_{\text{soil\_plant MD}}$) in the growing season 2018 from the rainfed (P2) and irrigated (P3) plots of the stony soil (F1) and silty soil (F2). Error bars indicate the standard deviation of the different values taken around midday (11 AM, 12AM, 1PM, and 2 PM) Leaf water potential and stomatal conductance were 2 sunlit leaves and one shaded leaf at each measured hour. Whole soil-plant hydraulic conductance was shown from 3 July when sap flow was measured. The black arrows indicates the irrigation events for the irrigated treatments F1P3 and F2P3 while the orange arrow indicates the irrigation application for the rainfed plot at the stony soil (F1P2).



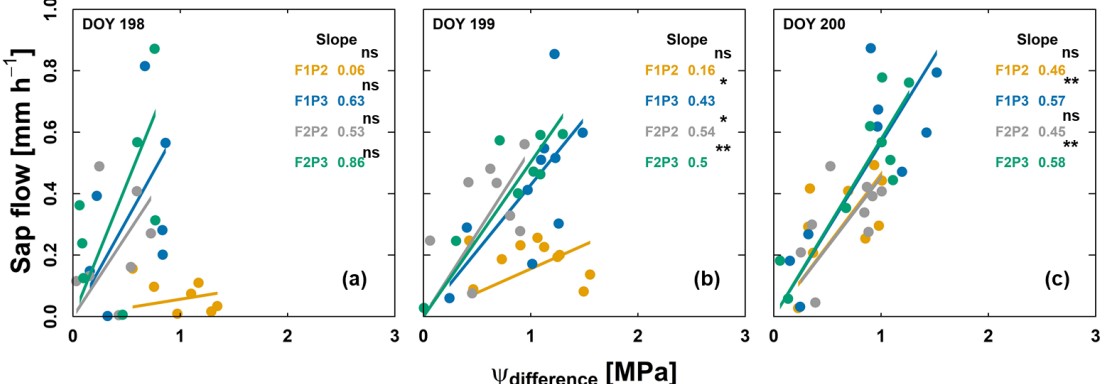

Figure 10: Relationship of sap flow and difference of effective soil water potential and sunlit leaf water potential ($\psi_{difference}$) from the rainfed (P2) and irrigated (P3) plots of the stony soil (F1) and silty soil (F2) on three consecutive measurement days from predawn in 2018 (a) DOY 198, (b) DOY 199 and (c) DOY 200. Crop was irrigated on 18 July (DOY 199) at 1 PM, 1 PM, and 4 PM for F1P3, F2P3, and F1P2, respectively (22.75 mm for each plot). The unit of slope in the linear regression (or soil-plant hydraulic conductance) is mm h$^{-1}$ MPa$^{-1}$. Regression was based on the DEMING approach. The asterisk which are next to the slopes indicate a significant correlation between two variables according to Pearson method (ns: non-significant; * p < 0.05; ** p < 0.01; *** p < 0.001).





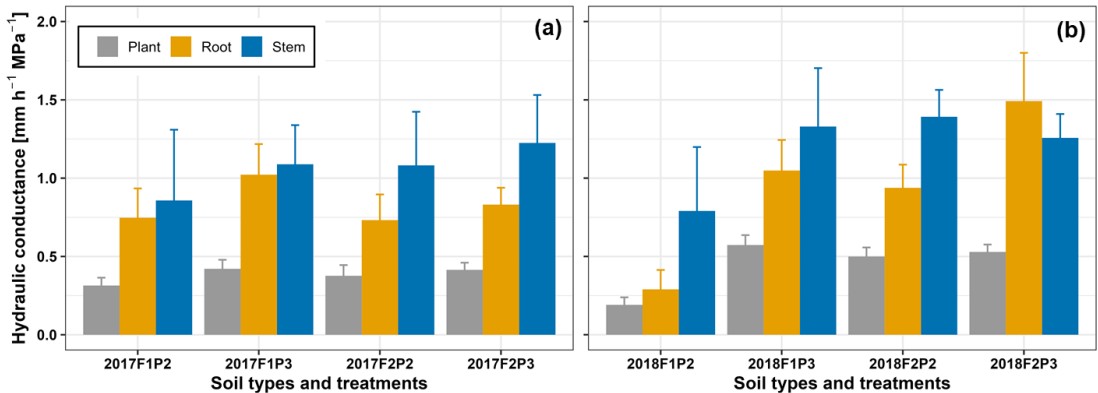

Figure 11: Comparison of different midday hydraulic components (mm h$^{-1}$ MPa$^{-1}$): soil-plant (grey bars), soil-root (yellow bars), and stem (blue bars) from the rainfed (P2) and irrigated (P3) plots of the stony soil (F1) and silty soil (F2) in the two growing seasons (a) in 2017 and (b) in 2018. The error bars indicate the standard deviation from measurements around midday (11 AM, 12AM, 1PM, and 2 PM) in different measured days (in 2017 with n = 4 x 9 days, Supplementary material 6, 7, and Fig. 8 and in 2018 with n = 4 x 10 days, Supplementary material 6, 8, and Fig. 9).



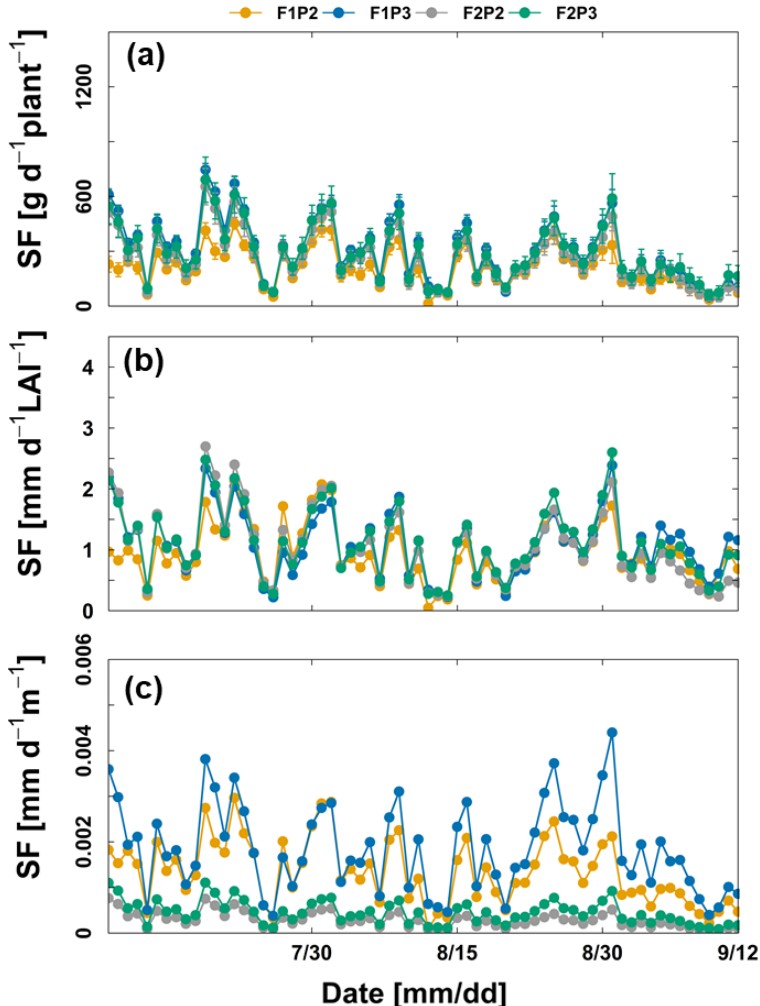

Figure 12: Comparison of sap flow (SF) in growing season 2017 from the rainfed (P2) and irrigated (P3) plots of the stony soil (F1) and silty soil (F2) with (a) sap flow per single plant (b) sap flow per leaf area index (LAI) and (c) sap flow per total root length. Data is shown from 9 July to 12 September 2017. Error bars in (a) indicate the standard deviation of the sap flow measurements in the five different maize plants.



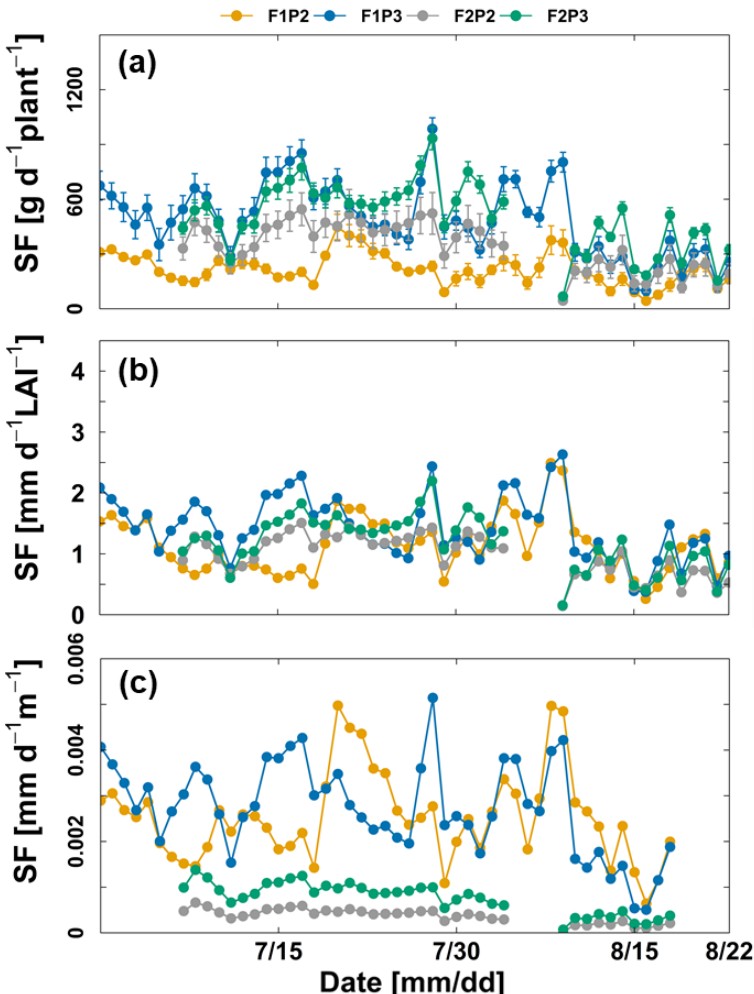

Figure 13: Comparison of sap flow (SF) in growing season 2018 from the rainfed (P2) and irrigated (P3) plots of the stony soil (F1) and silty soil (F2) with (a) sap flow per single plant (b) sap flow per leaf area index (LAI) and (c) sap flow per total root length. Data is shown in (a, b) from 29 June and 6 July for the stony soil (F1) and silty soil (F2), respectively to 21 August, 2018. Missing values of the beginning of the growing season and from 3 August to 6 August 2018 in the F2P2 and F2P3 were due to the missing values of measured sap flow because of sensor disconnection. Missing values in (c) at the end of the growing season in F2P2 and F2P3 was due to no availability of root measurement. Error bars in (a) indicate the standard deviation of the sap flow measurements in the five different maize plants.





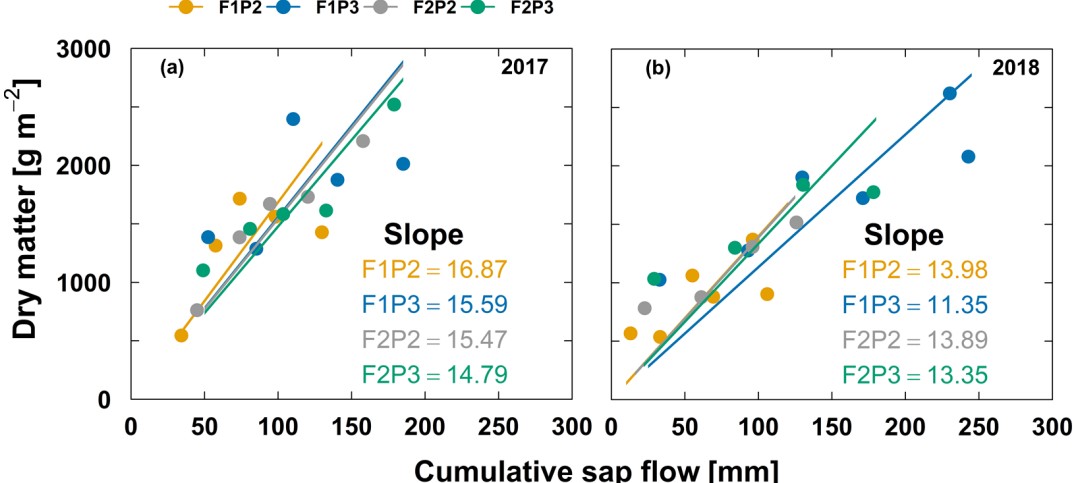

Figure 14: Relationship of aboveground dry matter and cumulative sap flow from the rainfed (P2) and irrigated (P3) plots of the stony soil (F1) and silty soil (F2) in the two growing seasons (a) 2017 and (b) 2018. The unit of slope linear relationship is g mm$^{-1}$. The less number of data points in (b) in 2018 from the F2P2 and F2P3 plots were due to the missing values of measured sap flow because of sensor disconnection.



## 1 Acknowledgements

This work has partially been funded by Federal Ministry of Education and Research (BMBF through European SUSCAP project – 031B0170B). We acknowledge the support by the SFB/TR32 "Pattern in Soil–Vegetation–Atmosphere Systems: Monitoring, Modelling, and Data Assimilation" funded by the Deutsche Forschungsgemeinschaft (DFG). Thuy Nguyen and Thomas Gaiser also thank the DETECT – CRC 1502 research program which is funded by DFG. We thank Dr. Matthias Langensiepen for his supports and technical help in the TR32 project. We would like to thank all the student assistants and technicians for their considerable efforts to collect the data in the field and the laboratories.



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

**Author contribution**

Huu Thuy Nguyen, Thomas Gaiser, Jan Vanderborght, and Frank Ewert: Conceptualization; Huu Thuy Nguyen, and Hubert Hüging: Data curation and data quality check (aboveground measurements); Lena Lärm, Felix Bauer, Anja Klotzsche, Jan Vanderborght, and Andrea Schnepf: data curation and data quality check (belowground measurements); Huu Thuy Nguyen: Formal data analysis and visualization; Thomas Gaiser, Jan Vanderborght, Andrea Schnepf, and Frank Ewert: Funding acquisition & Project administration; Huu Thuy Nguyen: writing – original draft; all authors: review, editing, and finalizing the manuscript.

**Competing interests**

This manuscript has not been published and is not under consideration for publication in any other journal. All authors agreed and approved the manuscript and its submission to this journal. We declare there is no conflict of interest.

**Code/Data availability**

The meteorological data were collected from a weather station in Selhausen (Germany) which belongs to the TERENO network of terrestrial observatories. Weather data are freely available from the TERENO data portal (https://www.tereno.net/ddp/dispatch?searchparams=freetext-Selhausen, last access: October 2020) (TERENO, 2020). The data which were obtained from the minirhizotron facilities (under- and aboveground) are available from the corresponding author on reasonable requests.