# Peer review of "contrasting vapor pressure deficit"

_EGUsphere, 2023_

## Author Comment (AC2)

**Referee #1: Dr. Jos C van Dam**

Dear Dr. van Dam,
Thank you very much for your constructive comments and advice that will help to improve the manuscript and the study. Please kindly follow the point-to-point responses (in blue color) to your comments (in black color) below. Since the revised MS was not prepared at this stage, the referred line numbers here were from the originally submitted MS. All detailed changes and revision were listed here.
On behalf of the authors
Thuy Nguyen

**General comments**

The authors performed an extensive study on maize growth at two distinct soil textures with different water regimes for a normal and extremely dry weather year. During the growing season, they collected a comprehensive dataset on root length, soil water potential, sap flow, leaf water potential, leaf transpiration, stomatal conductance, net photosynthesis and maize growth observations. These data were used to derive hydraulic conductances for the soil-root system, the plant stems and the entire plant system, to describe the maize reaction to dry weather and soil conditions and to determine the maize water use efficiency.

The paper addresses very relevant scientific questions related to the growth and management of one of the most important staple crops. The methods are clearly described and valid. The experimental results are well presented in 14 informative graphs and supplementary material and are objectively in detail described. The discussion is systematic and justified. Both the introduction and discussion are embedded in current scientific literature. The conclusions are sound and very useful for state-of-the-art crop-soil modeling.

In general, the used language is fluent and precise. Below I listed some minor text errors.

Thank you very much for your supportive summary and comments on the presented data, results, and findings of the study. Our responses to the detailed comments were below:

**Specific comments**

Line 66: Omit "of"

We delete "of" between "analyzed" and "genetic" words.

Line 117: Add space in "genotypesduring"

We added space to separate this typo. Thanks. See also similar comment from Referee#2

Line 134: "it" should be "it's"

We corrected this typo. See also similar comment from Referee#2

Line 178: "and 2018F1P3" should be "and 2018F2P3"

Many thanks. We corrected it. It will be "2018F1P3 and 2018 F2P3"

Line 180: omit "in"

We deleted "in".

Line 187: "form" should be "from"

We corrected the typo. Thanks.

Line 593-596: rewrite sentence

We revised the mentioned sentence. Here is the new sentence: "Maize plants had lower plant hydraulic conductance and more negative soil water potential in the rainfed plot in stony soil and they exhibited earlier stomatal closure as compared to the same plot in the silty soil"

Line 649: "effects" should be "effect"

We corrected the typo.

Line 650: "which based" should be "which was based"

We corrected the typo.

Line 667-669: rewrite sentence

We rewrote the mentioned sentence. Here is the new sentence: "In addition to stomatal regulation, leaf growth and plant size adjustments are important to regulate the transpiration and water use efficiency in the same year."

---

## Author Comment (AC3)

**Referee#2:**

Dear Referee#2,
Thank you very much for your constructive comments and advice that will help to improve the manuscript and the study. Please kindly follow the point-to-point responses (in blue color) to your comments (in black color) below. Since the revised MS was not prepared at this stage, the referred line numbers here were from the originally submitted MS. All detailed changes and revision were listed here.
On behalf of the authors
Thuy Nguyen

Review of the paper « **Responses of field-grown maize to different soil types, water regimes, and contrasting vapor pressure deficit** «

The paper reports the results of two experiments in which a maize genotype is grown under two soil types, i.e. a stony soil type and a silty soil type, and under two water treatment, i.e. a rainfed treatment that suffers water stress and a fully irrigated treatment. The authors then perform a number of measurement on the stomatal conductance, transpiration, sap flow rate, and assimilation rate, together with a number of water potential measurements at different levels. They also measured the root:shoot ratio in this factorial. One of the important results is the large variation between soil, but also between water treatment, in the root:shoot ratio. This has consequences for the hydraulic conductance of the soil-root-leaf-atmosphere continuum, where the main result is the demonstration that root system and soil-plant hydraulic conductance depended strongly on the soil texture. This is an important result that confirms other recent findings.

The results of the paper are very important, especially in showing the important role of the soil in shaping up plant physiological responses related to the plant hydraulic properties, and as such need to be published.

Many thanks for your supportive summary and comments on the data, results and findings of the study.

That said, I find that the paper is quite long and with too many figures (14). It has a discussion that I think is too long (11 pages) and contains complex elements of the results and does not make enough effort of summarizing, part by part, what are the key messages that need to come up. Therefore, I think the papers needs to be made more concise, may be less technical in some parts of it, before it can be published. It gives at times the impression that it was written as a long report about a series of results, lacking in focus sometimes. As it stands, it does not pay justice to the results that are presented and to the work that was done.

Thank you very much for your constructive and helpful comments. As suggested, we shortened the discussion and removed the information which was repeated from the result section e.g. see the response to comments on L435-478 (shorten the section of root discussion). Also, we have revised the text and added the key messages. Please see the responses to your comments on L389-399 (improvement and clarification in text for Figure 11), L482-489 (add interpretation and conclusion), and L482-522 (add conclusion). These revisions make the MS more concise and less technical in the mentioned sections. Since the combination of Figure 4, 5 and 6 will result in a large Figure, we still keep the number of figures as they are. This will compromise with the comment from Reviewer#1 who supported to keep 14 figures (also supplementary materials) which were informative and relevant to draw the conclusion.

Minor comments:

Thank you very much for your detailed comments and suggestions. We responded to your comments as following:

L108: Replace 'that' by 'because they'

This is done. We replaced with these suggested words.

L117: Space

This is done. See also comments from Referee#1

L128: Insert '.' After 'conductance'

This is done. Thank you.

L134: replace 'it' by 'its'

This is done. See also the comment from Referee#1.

L138: replace 'includes' by 'including'

This is done.

L146-147: may be replace 'different components (root, stem and whole soil-plant hydraulic conductance)' by 'different components of the hydraulic conductance (root, stem, whole soil-plant)'

Thank you for the suggestion. We replaced.

L155-157: Does each of these "rhizitrone facilities" represent what you call later "a site"??

We now consistently used the term "rhizotrone facilities" and removed the "site". More specifically we changed it in the line 158, line 175, and line 478.

L187: replace 'form' by 'from'

This is done. See also comment from Referee#1.

L198: Do you mean to say that over the 7m length of each of these 54 horizontal tubes, 20 measurements (images) were taken in each of them??

Yes. This was what we mean. In total, we have 20 x 2 (left and right sides) = 40 images from one tube. We rephrased the sentence. Here is the new sentence: "Root images were taken at 20 fixed positions from the left- and right-hand sides of each tube weekly (or biweekly) during the growing seasons."

L206: replace 'of' by 'in'

This was revised.

L210-211: Do you mean to say that you harvested five subplots, each of a 1m2, in each replication?? If that is the case, how did you treat the data? You averaged them?

At harvest, we were able to collect biomass samples with 5 replications (each of 1 m$^2$). As the aboveground dry matter data was shown in Figure 14 and in Supplementary material Figures 3a and 3b, the point at harvest was the average of 5 replications. We added one short sentence for the caption of each figure to clarify how we treated the dry matter data at harvest. Now as: "For aboveground dry matter, each point represents the average of two sampling replicates, except the harvest with 5 sampling replicates."

L304-305: The question is whether you measured such root cracks. And also what was the size of these putative crack as, if too large, roots may not be able to grow through such air-filled gaps. So, may be clarify whether you had any kind of indication of such cracks, and about their size (in Vertisol you could have cracks that would be 10 cm wide and surely no roots would go through that, so, it is important to know more about soil cracks in that silty soil).

Thank you very much for your suggestion. Your suggestion and concerns are also relevant to the comments from Community (Community comments – Dr. Oliver Dilly) on this manuscript regarding the definition of the soil types. You are right that with the special soils like Vertisols, the cracks could be up to 10 cm wide and roots will suffer from air-filled gaps. For the field in our study, the dominant soil is a Haplic Luvisol that contains a layer with clay accumulation (silty loam texture) (Weihermüller et al., 2007). The thickness of the silty loam layer varies strongly along the slope of the field. It is up to 3 m thick at the bottom of the slope and not present at the top. One rhizotron facility is located at the top (stony) and has a stone content of >60%. The second rhizotron facility is located at the bottom of slop. It has a thick layer with silty loam and a negligible stone content (<4%) and it is characterized by deep soil cracks when dry. However, the cracks were not large (< 5 mm wide) (Morandage et al., 2021). Thus, with this size of the cracks, roots were still able to grow.

References:

Morandage, S., J. Vanderborght, M. Zörner, G. Cai, D. Leitner, et al. 2021. Root architecture development in stony soils. Vadose Zo. J. (April): 1–17. doi: 10.1002/vzj2.20133.

Weihermüller, L., Huisman, J. A., Lambot, S., Herbst, M., & Vereecken, H. (2007). Mapping the spatial variation of soil water content at the field scale with different ground penetrating radar techniques. Journal of Hydrology, 340, 205–216. https://doi.org/10.1016/j.jhydrol.2007. 04.013

L312-328 and Figs 4-6: I wonder if you should group the three figures by showing the continuation of these 4 types of measurements on the same graph, indicating when irrigation would begin and stop. It would much better allow to compare different days, and compare soil and soil-by-water treatment combinations among them for these different factors. The way the figures are laid out and given the relatively small differences it is a bit difficult to do that.

Thank you very much for your suggestion. We understood your point that putting three figures together in one graph will enhance the comparison for different days, before and after irrigation. However, we think combination of all figures (17 sub-plots showing hourly values of 4 main variables – LWP, E, Gs, and An with PAR and VPD for two soil types and two water treatments) could end up with a large and complex figure. Thus, we would like to keep the figures as they are now. However, we added black arrows which indicate when irrigation was applied (e.g. Figure 5n, 5o and 5q). Moreover, the time of irrigation is now mentioned in the text and the captions. We hope the three separated figures with the high quality of resolution and medium information will convey the differences of An, Gs, LWP, and E for different measurement days.

Fig5

[Figure]

L377-378: Here you switch to DOY, whereas in earlier figures you used the calendar dates. This may be a detail but the data presented are sufficiently complex that it would be good to harmonize what can be. So either choose DOY or calendar dates but stick to it.

Thank you very much for your comment and suggestion. We now use consistently the calendar dates in the text in describing the results and discussion that were related to Figure 3 and Figure 10. More specifically:

Line 302 (with Figure 3): add date, now as "Moreover, total root length was relatively equal among treatments at the start of stem elongation (8 June - DOY 159) in both years, while this was the opposite for the ratio of root length to shoot dry matter."

Line 375-387 (with Figure 10): use date, now as "The slope of linear relationship between sap flow and difference of $\psi_{soil\_effec}$ and $\psi_{sunlitleaf}$ is shown for three consecutive days (leaf water potential measurements from the predawn) and before and after irrigation applications (17, 18, and 19 July 2018) (Figure 10). On both dates 17 and 18 July, the difference between $\psi_{soil\_effec}$ and $\psi_{sunlitleaf}$ was around -1.6 MPa with very low transpiration rates in the treatment F1P2 which was associated with very low plant hydraulic conductance and leaf curling. The whole plant hydraulic conductance was disrupted on these two days (0.06 and 0.16 mm $h^{-1}$ $MPa^{-1}$ for 17 and 18 July, respectively). Water was supplied on 18 July at 1 PM for the irrigated plots (F1P3, F2P3) as well as F1P2 at 4 PM (for saving plant from death due to severe drought stress). $K_{soil\_plant}$ was slightly changed (0.43 and 0.57 mm $h^{-1}$ $MPa^{-1}$ for F1P3 on 18 and 19 July, respectively and 0.5 and 0.58 mm $h^{-1}$ $MPa^{-1}$ for F2P3 on 18 and 19 July, respectively). However, the increase of $K_{soil\_plant}$ was substantial in the F1P2 after the irrigation. Soil water replenishment and an increase in the root - soil contact (Fig. 9a) allowed the $K_{soil\_plant}$ to recover overnight to 0.46 mm $h^{-1}$ $MPa^{-1}$. This resulted in a narrower water potential gradient between root zone and sunlit leaf and in a higher transpiration rate on 19 July."

Line 463-464 (with Figure 3): add date, now as: "A slightly higher root: shoot ratio in the F1P2 treatment compared to F1P3 (13 July - DOY 194 & 12 September – DOY 255) was observed in 2017 while the root: shoot ratio in the two treatments was almost the same on 18 July - DOY 199 and 16 August – DOY 228 in 2018 (Fig. 3)."

Line 588 (with Figure 10): add date, now as "The $K_{soil\_plant}$ slightly increased after irrigation (18 July - DOY 199 in Fig. 10b) corresponding with the smaller $\psi_{difference}$ (Fig. 10b) and an increase in stomatal conductance (Fig. 9c)."

Also, x axis in Figure 8d and Figure 9d will be changed, from "DOY" to "Date".

[Figure]

[Figure]

L389-399: I see the graph of Fig 11, but the end of the paragraph makes an important interpretation on the data that I have difficulties to follow, possibly because the data that are presented are not adequate for the reader to easily understand that interpretation. Could you clarify?

Many thanks for your suggestion. We revised the text within and at the end of the mentioned paragraph to make them clearer. Now as:

"Seasonal average of different midday hydraulic conductance components (root system hydraulic conductance - $K_{soil\_root}$, stem hydraulic conductance – $K_{stem}$, and whole plant hydraulic conductance –

$K_{soil\_plant}$) are shown in Figure 11. In the same year, the $K_{stem}$ was not much different among F1P3, F2P2, and F2P3 plots. The $K_{stem}$ of those plots was slightly higher than in the F1P2 in both years. In general, the $K_{soil\_root}$ was lower than the $K_{stem}$. Overall, the estimated $K_{soil\_plant}$ was around 1/ (1/$K_{soil\_root}$ +1/$K_{stem}$) regardless of soil types, years, and water treatments. The $K_{soil\_root}$ and $K_{soil\_plant}$ in the F1P2 in 2018 was much lower than the remaining plots while the $K_{soil\_root}$ and $K_{soil\_plant}$ were not much different among plots in 2017. Our results indicated that there was an impact of soil hydraulic conductance on $K_{soil\_root}$ and $K_{soil\_plant}$. Although there is a large difference in total root length between the two soil types (e.g. F1P3 versus F2P2 or F2P3 versus F2P2), $K_{soil\_root}$ and $K_{soil\_plant}$ in those two plots were not much different. This could be explained by the fact that $K_{soil\_plant}$ was not only depended on root length but also depended on the variability of root segment hydraulic conductance. "

L440: Instead of saying that rooting depth was sensitive to the presence of crack, it would be clearer to say if presence of cracks increased/decreased rooting depth.

Many thanks to your suggestion. We revised the sentence. Now as: "Also, both simulations and observations indicated that rooting depth was increased due to the presence of cracks in the lower minirhizontron facility (Morandage et al., 2021) which could explain the high root length between 40 and 120 cm soil depths which was observed in the silty soil in both years."

L455: you mention root:shoot ratio but then talk about 200-1000 cmg-1, which is a density and not a ratio, please correct.

Thank you for your suggestion. We have used the term: "ratios of root length to shoot biomass" and have corrected the term throughout the text, as in the lines below:

Line 455: now as: "In terms of the ratios of root length to shoot biomass, our observations were in line with those reported in the same soil types for wheat in Cai et al., (2018)".

Line 458: now as: "Jorda et al., (2022) reported a wide range of ratios of root length to shoot biomass from 200 to 1000 cm g$^{-1}$ around flowering time of maize depending on the wild type and root hair mutant genotypes growing on either loamy or sandy soils."

Line 463: now as "We only observed much higher ratios of root length to shoot biomass in the rainfed plot (F2P2) as compared to the irrigated plot on the silty soil (F2P3).

Line 474: now as "The larger ratios of root length to shoot biomass in this F2P2 plot in 2018 as compared to F2P3 could be explained by the change of source: sink relations where more assimilates were devoted to root growth, even at a later growth stage."

Line 655-656 (in Conclusion): now as "Our results confirmed that root length and ratios of root length to shoot biomass were modulated by soil types and water treatment but less by seasonal evaporative demand. Increase ratio of root length to shoot biomass has been an important response of maize that allows plants to extract more water under drought stress that occurred rather in the silty soil but less in the stony soil due to the higher content of stony material."

L435-478: This part of the discussion is too long, repeats a number of result part, and then brings elements of literature in a rather scattered and unrelated way (for instance talking about the soil nutrient effects on root length). And finally, it does not give a tangible conclusion of what we should retain about these root length differences between soil texture and between water treatments.

Many thanks for your suggestion. We revised and shortened the text by removing the repetition of result part (Line 463-466) and the soil nutrient effects on root length (Line 444-448), and ratio of root to leaf area (Line 466-469). Also, we removed the comparison of wheat (from previous studies, line 445-447 and line 452-456) and maize (our study) to make the discussion more focused on maize under different soils. In total, **we removed 17 sentences**. As suggested, we added one short sentence at the end of this section to conclude the root length differences were due to soil texture and soil water availability.

The original L435-478 is now much shorter, as:

"Our root observations showed that soil type considerably affected root growth more than water treatment (Figure 2). Root growth was strongly inhibited by the stony soil where much lower root length was observed than in the silty soil, especially in the deeper soil layers. This was consistent with the findings reported in (Morandage et al., 2021) where a linear increase of stone content resulted in a linear decrease of rooting depth across all stone contents and developmental stages. Also, both simulations and observations indicated that rooting depth was increased to the presence of cracks in the lower minirhizontron facility (Morandage et al., 2021) which could explain the high root length between 40 and 120 cm soil depths which was observed in the silty soil in both years.

In terms of the ratios of root length to shoot biomass, Ordóñez et al., (2020) has reported much larger figures of for instance 880 cm $g^{-1}$ in different locations and under different N application rates in maize growing in the Midwest of US. Jorda et al., (2022) reported a wide range of ratios of root length to shoot biomass from 200 to 1000 cm $g^{-1}$ around flowering time of maize depending on the wild type and root hair mutant genotypes growing on either loamy or sandy soils. More roots and higher ratios of root length to shoot biomass were found in the sand than in the loam in both wild type and root hair mutant genotypes (Jorda et al., 2022; Vetterlein et al., 2022). Cai et al., (2018) observed much larger ratios of root length to shoot biomass in drought stressed plots than in irrigated plot in both soil types in winter wheat which indicated the alternation of sink: source relationships to cope with water stress. This study emphasized that more assimilates are used to promote root growth and extract more water under drought stress. However, this was not the case for the stony soil in our work where the drought stress was more pronounced, especially in 2018. A drop of soil water potential (Supplementary material 2b), thus effective soil water potential (Figure 8a) was substantial from 10th July 2018 toward the harvest in the rainfed plot in the silty soil (F2P2) which was consistent with the reduction of leaf water potential (Fig. 8b), leaf area (Supplementary material 3c), total dry matter (Supplementary material 3d), and crop height (Supplementary material 4b) as compared the irrigated plot (F2P3). This indicates a mild water stress in 2018 in the rainfed plots on the silty soil. The larger ratios of root length to shoot biomass in this F2P2 plot in 2018 as compared to F2P3 could be explained by the change of source: sink relations where more assimilates were devoted to root growth, even at a later growth stage. Moreover, the low stone content and soil cracks (Morandage et al., 2021) might favor root growth in the deeper soil layers which are close to the shallow soil water table in the rhizotrone facility with silty soil (Vanderborght et al., 2010). In conclusion, both soil texture and water conditions influenced the root growth, however, effects of the former on root length was more pronounced than the latter."

L481: as affected by?

Thank you. We corrected the heading 4.2.1. Now as: "Leaf water potential and stomatal conductance as affected by soil water conditions"

L484: replace 'lower' by 'less'

We revised the sentence as suggested. Now as: "Moreover, the low stone content and soil cracks (Morandage et al., 2021) might favor root growth in the deeper soil layers which are close to the shallow soil water table in the rhizotrone facility with silty soil (Vanderborght et al., 2010)."

L482-489: you comment your results, then earlier results from others. And these are different. But you leave the reader hanging here: what is your interpretation/explanation for that?

Thank you. We revised the text: specifically moving the first sentence to the end of paragraph then adding one sentence for the explanation of impacts of soil on leaf water potential and stomatal conductance for the previous sentence. This will help the reader to have a clear message. The L482-489 now as:

"In the previous work, Koehler et al., (2022) reported that maize stomata closed at lower negative leaf water potentials in sand than in loam growing under controlled environment. Cai et al., (2022b) investigated transpiration response of pot-grown maize in two contrasting soil textures (sand and loam) and exposed to two consecutive VPD levels (1.8 and 2.8 kPa). Transpiration rate decreased at less negative soil matric potential in sand than in loam at both VPD levels. In sand, high VPD generated a steeper drop in stomatal conductance with decreasing leaf water potential which indicated that the transpiration and stomatal responses depend on soil hydraulics. In our study, stomata closed earlier and at more negative soil and leaf water potentials in the stony soil than in the silty soil (see Fig. 4, 5, 6 and 7). The lower soil water holding capacity of the stony soil compared to the silty soil resulted in lower soil water potential and smaller total plant hydraulic conductance which in turn led to earlier stomatal closure and to more negative soil water potential in the stony soil. "

L482-522: Here also is a fairly long piece of complex discussion that would need to end with a kind of message. I guess the message is that while Welcker and other have show genotypic differences earlier, here you show the soil influence on minimum psi Leaf.

Many thanks for your comment and suggestion. As suggested, we add one short sentence to conclude this key message. Now as: "In conclusion, our results confirmed that the minimum $\psi_{leaf}$ not only depended on genotypic differences but also was influenced by soil types and soil hydraulic conductance."

L525: Stomatal conductance?

This was revised.

L528: replace 'estimate' by 'estimation'

This was done. Thanks.

L533-539: Here I think it would be better to stick to the units of mm h-1 MPa-1 and then convert the values in 10-5 h-1 from the citations into that mm h-1 MPa-1 unit which has been used in your graphs.

Thanks. We changed the unit in $10^{-5}$ $h^{-1}$ into mm $h^{-1}$ $MPa^{-1}$ throughout the text from L533-539.

L540-541: That sentence is not understandable

We removed the L540-543 since we think it is also not relevant for the context.

L649: contrasting

It was revised.

L666: …. and decreased at more negative…

It was revised. Now as: "The stomata conductance was smaller and decreased at more negative leaf water potentials in stony soil than in silty soil."

L667: remove 'plant'

Many thanks. The redundant word was removed.

L668: replace 'that' by 'and'

This was done. See also comment from Referre#1.

---

## Author Response (AR2)

Dear Dr. Vico (handling editor) and Dr. Rammig (co-editor-in-chef)

Thank you very much for your handling and support for the evaluation process. Also, thank you for email and request of major revision of the manuscript. To check the changes of MS, please kindly see the version with track changes and clean/final version. We responded to point to point comment and question from the handling editor and two reviewers (which have been done and shown in the open discussion process). Here, the line numbers in the below responses are **referred to the track changes version of the MS.**

On behalf of the authors

Thuy Nguyen

**Comments from the handling editor**

I concur with the evaluation of the two reviewers that the manuscript reports novel and interesting data, but needs some improvements.

Many thanks for your supportive comments on the MS.

I urge the authors to pay particular attention to the aspect of conciseness, not just per se but to have the main points emerge. This means thoroughly revisit the discussion section, critically assessing which are the main points and focusing on those, removing/reducing the prevalence of the others. This also means critically assessing the need for all the figures now in the text. The end goal is not so much reducing the number of figures by combining panels in more complex ones or shortening the text, but selecting for the important material, and, if deemed necessary, place some material in the SI, thus allowing for a read at different depths.

Many thanks for your suggestions on the content in the discussion section, related main points, and remove of other information as well as the number of figures.

Your comments are in line with the reviewer#2. The discussion should be more concise and focused which the key points should be brought up and text should be reduced. We have done substantial reduction of text and added the key points.

In the section 4. 1 in the discussion, line 459-465 (discussion about other studies) were removed. Line 466-470 and 481-488 were also removed. Key point at the end of section was added (line 498-499, see also comment of Reviewer#2)

In the section 4.2.1 in the discussion, line 521-548 were removed (see also your below comment). One sentence was added to conclude the results (line 549-550).

In the section 4.2.2 line 569-572 was removed (see also comment from Reviewer#2).

In the section 4.2.2 in the discussion, line 574-578 and 593-598 were removed (removing/reducing the prevalence of the others).

In the section 4.2.3 in the discussion, line 602-611 were removed (avoid repetition of the result) and line 611-614 were removed (removing/reducing the prevalence of the others, avoid the repetition with line 631-635)

In the section 4.3 in the discussion, line 666-671 were removed (reduce the prevalence of the others).

For the figures, as suggested, we selected and removed the Figure 4, Figure 6, Figure 11, and Figure 12 to Supplementary materials. The Figure and supplementary materials were revised and updated throughout the main text. Ultimately, there were 10 key figures and one table that were belonged to the main text. From our thought, this made the result and discussion sections shorter and more focused. This is also in line with suggestions from the reviewer#2.

All dates are expressed as such or days of the year, but with no reference to the developmental stage of the plants on those days. It is important to clarify those or, at least explain whether the different treatments caused the plants to be in a different developmental stage on the same date.

Many thanks for your comments. We observed difference in emerge, tasseling and silking dates for two growing seasons due to the differences of sowing dates and temperature. But, we did not observe the difference among water treatments and soil types within one season. We added one sentence to clarify the phenology information (Line 210-214) and referred the developmental stages (line 241-243) maize of these two growing seasons in Nguyen et al., 2022a (see Table 1 in their study).

**Minor comments:**

L26: did the minimum leaf water potential occur at the same time and developmental stage?

The measurement was done in the same day and hour. The developmental stages were mentioned in the above responses. We revised the sentence and made it more precise:

"The seasonally observed minimum leaf water potential ($\psi_{leaf}$) varied from around -1.5 MPa in the rainfed plot in 2017 to around -2.5 MPa in the same plot of the stony soil in 2018."

L63: 'was' but it is no longer so? I suggest using 'is'

Many thanks. It was corrected.

L159. Does this mean that the irrigated treatment was not replicated?

Yes. The construction of cellar for minirhizotubes were complex, originally, there were only one irrigated, one rainfed, and one sheltered plot in each rhizotrone facility. From 2017 onwards, due to the collapse of rain-out shelter, there were one irrigated plot and two rainfed plots in each soil type.

L170: how was the irrigation interval selected?

We have described in Nguyen et al., (2022a). The rainfall amount was manually recorded by the rain gage close to the field. Irrigation water applications were determined as the biweekly sum of the calculated crop evapotranspiration (ETc). The daily ETc was estimated based on reference evapotranspiration (ETo) using Penman-Monteith equation and single crop coefficient (Kc) for winter wheat and maize (Allen et al., 1998). We added one sentence after line 182. "Detailed estimates of irrigation amount and intervals could be found in Nguyen et al., (2022a)."

**Additional private note (visible to authors and reviewers only):**

The plagiarism check, which is run on each submission, evidenced some substantial similarities with Nguyen et al 2022 Ag Forest Met. While the content of this manuscript is different, copy pasting is never acceptable, even from own works and even in the methods. Yet, I see large overlaps in two paragraphs of the intro, parts of the methods (in particular Sections 2.3.2 and 2.3.3), L510-522 and Fig 4 captions. Please remove that completely.

Many thanks for your comments and the advice. We have removed completely the repeated text and information as suggested.

For the introduction: two paragraphs, specifically line 65-69 and line 130-133 were removed.

For the materials and methods sections, we removed the line 215 -220 (in the section 2.3.2), line 223-225, 227-228, line 230-232, and line 236-239 (in the section 2.3.3). The remaining text of those two sections were combined into one section "2.3.2. Crop growth, leaf gas exchange, leaf water potential and sap flow measurement". We referred the readers to the detailed measurement from Nguyen et al., 2022a and Nguyen et al., 2020 (in line 241-243).

Line 521-548 in the discussion were removed as suggested. This revision was coincident with the comment from the review#2 for line 482-522 that the lengthy discussion should be shortened.

In addition we added one more reference (line 206 and 871) for the used root data and updated the acknowledgement (line 739-740). We removed the reference in 902-904

**Referee #1: Dr. Jos C van Dam**

Dear Dr. van Dam,
Thank you very much for your constructive comments and advice that will help to improve the manuscript and the study. Please kindly follow the point-to-point responses (in blue color) to your comments (in black color) below. The line numbers mentioned in the responses below are referred to the track changes version of the MS. All detailed changes and revision were listed here.
On behalf of the authors
Thuy Nguyen

**General comments**

The authors performed an extensive study on maize growth at two distinct soil textures with different water regimes for a normal and extremely dry weather year. During the growing season, they collected a comprehensive dataset on root length, soil water potential, sap flow, leaf water potential, leaf transpiration, stomatal conductance, net photosynthesis and maize growth observations. These data were used to derive hydraulic conductances for the soil-root system, the plant stems and the entire plant system, to describe the maize reaction to dry weather and soil conditions and to determine the maize water use efficiency.

The paper addresses very relevant scientific questions related to the growth and management of one of the most important staple crops. The methods are clearly described and valid. The experimental results are well presented in 14 informative graphs and supplementary material and are objectively in detail described. The discussion is systematic and justified. Both the introduction and discussion are embedded in current scientific literature. The conclusions are sound and very useful for state-of-the-art crop-soil modeling.

In general, the used language is fluent and precise. Below I listed some minor text errors.

Thank you very much for your supportive summary and comments on the presented data, results, and findings of the study. Our responses to the detailed comments were below:

**Specific comments**

Line 66: Omit "of"

We delete "of" between "analyzed" and "genetic" words.

Line 117: Add space in "genotypesduring"

We added space to separate this typo. Thanks. See also similar comment from Referee#2

Line 134: "it" should be "it's"

We corrected this typo. See also similar comment from Referee#2

Line 178: "and 2018F1P3" should be "and 2018F2P3"

Many thanks. We corrected it. It will be "2018F1P3 and 2018 F2P3"

Line 180: omit "in"

We deleted "in".

Line 187: "form" should be "from"

We corrected the typo. Thanks.

Line 593-596: rewrite sentence

We revised the mentioned sentence. Here is the new sentence: "Maize plants had lower plant hydraulic conductance and more negative soil water potential in the rainfed plot in stony soil and they exhibited earlier stomatal closure as compared to the same plot in the silty soil"

Line 649: "effects" should be "effect"

We corrected the typo.

Line 650: "which based" should be "which was based"

We corrected the typo.

Line 667-669: rewrite sentence

We rewrote the mentioned sentence. Here is the new sentence: "In addition to stomatal regulation, leaf growth and plant size adjustments are important to regulate the transpiration and water use efficiency in the same year."

**Referee#2:**

Dear Referee#2,
Thank you very much for your constructive comments and advice that will help to improve the manuscript and the study. Please kindly follow the point-to-point responses (in blue color) to your comments (in black color) below. The line numbers mentioned in the responses below are referred to the track changes version of the MS. All detailed changes and revision were listed here.
On behalf of the authors
Thuy Nguyen

Review of the paper « **Responses of field-grown maize to different soil types, water regimes, and contrasting vapor pressure deficit** «

The paper reports the results of two experiments in which a maize genotype is grown under two soil types, i.e. a stony soil type and a silty soil type, and under two water treatment, i.e. a rainfed treatment that suffers water stress and a fully irrigated treatment. The authors then perform a number of measurement on the stomatal conductance, transpiration, sap flow rate, and assimilation rate, together with a number of water potential measurements at different levels. They also measured the root:shoot ratio in this factorial. One of the important results is the large variation between soil, but also between water treatment, in the root:shoot ratio. This has consequences for the hydraulic conductance of the soil-root-leaf-atmosphere continuum, where the main result is the demonstration that root system and soil-plant hydraulic conductance depended strongly on the soil texture. This is an important result that confirms other recent findings.

The results of the paper are very important, especially in showing the important role of the soil in shaping up plant physiological responses related to the plant hydraulic properties, and as such need to be published.

Many thanks for your supportive summary and comments on the data, results and findings of the study.

That said, I find that the paper is quite long and with too many figures (14). It has a discussion that I think is too long (11 pages) and contains complex elements of the results and does not make enough effort of summarizing, part by part, what are the key messages that need to come up. Therefore, I think the papers needs to be made more concise, may be less technical in some parts of it, before it can be published. It gives at times the impression that it was written as a long report about a series of results, lacking in focus sometimes. As it stands, it does not pay justice to the results that are presented and to the work that was done.

Thank you very much for your constructive and helpful comments. As suggested, we shortened the discussion and removed the information which was repeated from the result section e.g. see the response to comments on L435-478 (shorten the section of root discussion). Also, we have revised the text and added the key messages. Please see the responses to your comments on L389-399 (improvement and clarification in text for Figure 11 (now figure 9), L482-489 (add interpretation and conclusion), and L482-522 (add conclusion). We reduced four figures (4, 6, 12, and 13) and moved to Supplementary materials.

These revisions make the MS more concise and less technical in the mentioned sections. This is in line with the suggestions from the editor.

Minor comments:

Thank you very much for your detailed comments and suggestions. We responded to your comments as following:

L108: Replace 'that' by 'because they'

This is done. We replaced with these suggested words.

L117: Space

This is done. See also comments from Referee#1

L128: Insert '.' After 'conductance'

This is done. Thank you.

L134: replace 'it' by 'its'

This is done. See also the comment from Referee#1.

L138: replace 'includes' by 'including'

This is done.

L146-147: may be replace 'different components (root, stem and whole soil-plant hydraulic conductance)' by 'different components of the hydraulic conductance (root, stem, whole soil-plant)'

Thank you for the suggestion. We replaced.

L155-157: Does each of these "rhizitrone facilities" represent what you call later "a site"??

We now consistently used the term "rhizotrone facilities" and removed the "site". More specifically we changed it in the line 159, line 176, and line 497.

L187: replace 'form' by 'from'

This is done. See also comment from Referee#1.

L198: Do you mean to say that over the 7m length of each of these 54 horizontal tubes, 20 measurements (images) were taken in each of them??

Yes. This was what we mean. In total, we have 20 x 2 (left and right sides) = 40 images from one tube. We rephrased the sentence. Here is the new sentence: "Root images were taken at 20 fixed positions from the left- and right-hand sides of each tube weekly (or biweekly) during the growing seasons." We added the updated reference for the root data collection which was described in detail in the recent publication (line 206).

"Lärm, L., F.M. Bauer, N. Hermes, J. van der Kruk, H. Vereecken, et al. 2023. Multi-year belowground data of minirhizotron facilities in Selhausen. Sci. Data 10(1): 1–15. doi: 10.1038/s41597-023-02570-9.

L206: replace 'of' by 'in'

This was revised.

 L210-211: Do you mean to say that you harvested five subplots, each of a 1m2, in each replication?? If that is the case, how did you treat the data? You averaged them?

At harvest, we were able to collect biomass samples with 5 replications (each of 1 $m^2$). As the aboveground dry matter data was shown in Figure 14 (now figure 10) and in Supplementary material Figures 3a and 3b, the point at harvest was the average of 5 replications. We added one short sentence for the caption of each figure to clarify how we treated the dry matter data at harvest. Now as: "For aboveground dry matter, each point represents the average of two sampling replicates, except the harvest with 5 sampling replicates."

L304-305: The question is whether you measured such root cracks. And also what was the size of these putative crack as, if too large, roots may not be able to grow through such air-filled gaps. So, may be clarify whether you had any kind of indication of such cracks, and about their size (in Vertisol you could have cracks that would be 10 cm wide and surely no roots would go through that, so, it is important to know more about soil cracks in that silty soil).

Thank you very much for your suggestion. Your suggestion and concerns are also relevant to the comments from Community (Community comments – Dr. Oliver Dilly) on this manuscript regarding the definition of the soil types. You are right that with the special soils like Vertisols, the cracks could be up to 10 cm wide and roots will suffer from air-filled gaps. For the field in our study, the dominant soil is a Haplic Luvisol that contains a layer with clay accumulation (silty loam texture) (Weihermüller et al., 2007). The thickness of the silty loam layer varies strongly along the slope of the field. It is up to 3 m thick at the bottom of the slope and not present at the top. One rhizotron facility is located at the top (stony) and has a stone content of >60%. The second rhizotron facility is located at the bottom of slop. It has a thick layer with silty loam and a negligible stone content (<4%) and it is characterized by deep soil cracks when dry. However, the cracks were not large (< 5 mm wide) (Morandage et al., 2021). Thus, with this size of the cracks, roots were still able to grow.

References:

Morandage, S., J. Vanderborght, M. Zörner, G. Cai, D. Leitner, et al. 2021. Root architecture development in stony soils. Vadose Zo. J. (April): 1–17. doi: 10.1002/vzj2.20133.

Weihermüller, L., Huisman, J. A., Lambot, S., Herbst, M., & Vereecken, H. (2007). Mapping the spatial variation of soil water content at the field scale with different ground penetrating radar techniques. Journal of Hydrology, 340, 205–216. https://doi.org/10.1016/j.jhydrol.2007. 04.013

L312-328 and Figs 4-6: I wonder if you should group the three figures by showing the continuation of these 4 types of measurements on the same graph, indicating when irrigation would begin and stop. It would much better allow to compare different days, and compare soil and soil-by-water treatment combinations among them for these different factors. The way the figures are laid out and given the relatively small differences it is a bit difficult to do that.

Thank you very much for your suggestion. We understood your point that putting three figures together in one graph will enhance the comparison for different days, before and after irrigation. We added black arrows which indicate when irrigation was applied (e.g. Figure 5n, 5o and 5q, now in Figure 4n, 4o, and 4q). Moreover, the time of irrigation is now mentioned in the text and the captions. We now moved the figure 4 and figure 6 to the supplementary material. Also, two Figures 12 and 13 were move to the supplementary materials. These will reduce the number of figures in the main text but the results and information were still maintained.

Fig5 (now Fig 4)

[Figure]

L377-378: Here you switch to DOY, whereas in earlier figures you used the calendar dates. This may be a detail but the data presented are sufficiently complex that it would be good to harmonize what can be. So either choose DOY or calendar dates but stick to it.

Thank you very much for your comment and suggestion. We now use consistently the calendar dates in the text in describing the results and discussion that were related to Figure 3 and Figure 8 (now Figure 6). More specifically:

Line 302 (with Figure 3): add date, now as line 304: "Moreover, total root length was relatively equal among treatments at the start of stem elongation (8 June - DOY 159) in both years, while this was the opposite for the ratio of root length to shoot dry matter."

Line 379-390 (with Figure 10 now Figure 8): use date, now as "The slope of linear relationship between sap flow and difference of $\psi_{soil\_effec}$ and $\psi_{sunlitleaf}$ is shown for three consecutive days (leaf water potential measurements from the predawn) and before and after irrigation applications (17, 18, and 19 July 2018) (Figure 8). On both dates 17 and 18 July, the difference between $\psi_{soil\_effec}$ and $\psi_{sunlitleaf}$ was around -1.6 MPa with very low transpiration rates in the treatment F1P2 which was associated with very low plant hydraulic conductance and leaf curling. The whole plant hydraulic conductance was disrupted on these two days (0.06 and 0.16 mm $h^{-1}$ $MPa^{-1}$ for 17 and 18 July, respectively). Water was supplied on 18 July at 1 PM for the irrigated plots (F1P3, F2P3) as well as F1P2 at 4 PM (for saving plant from death due to severe drought stress). $K_{soil\_plant}$ was slightly changed (0.43 and 0.57 mm $h^{-1}$ $MPa^{-1}$ for F1P3 on 18 and 19 July, respectively and 0.5 and 0.58 mm $h^{-1}$ $MPa^{-1}$ for F2P3 on 18 and 19 July, respectively). However, the increase of $K_{soil\_plant}$ was substantial in the F1P2 after the irrigation. Soil water replenishment and an increase in the root - soil contact (Fig. 7a) allowed the $K_{soil\_plant}$ to recover overnight to 0.46 mm $h^{-1}$ $MPa^{-1}$. This resulted in a narrower water potential gradient between root zone and sunlit leaf and in a higher transpiration rate on 19 July."

Line 606-607 (with Figure 10 now Figure 8): add date, now as "The $K_{soil\_plant}$ slightly increased after irrigation (18 July - DOY 199 in Fig. 8b) corresponding with the smaller $\psi_{difference}$ (Fig. 8b) and an increase in stomatal conductance (Fig. 7c)."

Also, x axis in Figure 8d and Figure 9d will be changed, from "DOY" to "Date".

[Figure]

[Figure]

L389-399: I see the graph of Fig 11, but the end of the paragraph makes an important interpretation on the data that I have difficulties to follow, possibly because the data that are presented are not adequate for the reader to easily understand that interpretation. Could you clarify?

Many thanks for your suggestion. We revised the text within and at the end of the mentioned paragraph to make them clearer. Now in line 401-416 as:

"Seasonal average of different midday hydraulic conductance components (root system hydraulic conductance - $K_{soil\_root}$, stem hydraulic conductance – $K_{stem}$, and whole plant hydraulic conductance – $K_{soil\_plant}$) are shown in Figure 9. In the same year, the $K_{stem}$ was not much different among F1P3, F2P2, and F2P3 plots. The $K_{stem}$ of those plots was slightly higher than in the F1P2 in both years. In general, the $K_{soil\_root}$ was lower than the $K_{stem}$. Overall, the estimated $K_{soil\_plant}$ was around 1/ (1/$K_{soil\_root}$ +1/$K_{stem}$) regardless of soil types, years, and water treatments. The $K_{soil\_root}$ and $K_{soil\_plant}$ in the F1P2 in 2018 was much lower than the remaining plots while the $K_{soil\_root}$ and $K_{soil\_plant}$ were not much different among plots in 2017. Our results indicated that there was an impact of soil hydraulic conductance on $K_{soil\_root}$ and $K_{soil\_plant}$. Although there is a large difference in total root length between the two soil types (e.g. F1P3 versus F2P2 or F2P3 versus F2P2), $K_{soil\_root}$ and $K_{soil\_plant}$ in those two plots were not much different. This could be explained by the fact that $K_{soil\_plant}$ was not only depended on root length but also depended on the variability of root segment hydraulic conductance. "

L440: Instead of saying that rooting depth was sensitive to the presence of crack, it would be clearer to say if presence of cracks increased/decreased rooting depth.

Many thanks to your suggestion. We revised the sentence. Now line 457 as: "Also, both simulations and observations indicated that rooting depth was increased due to the presence of cracks in the lower minirhizontron facility (Morandage et al., 2021) which could explain the high root length between 40 and 120 cm soil depths which was observed in the silty soil in both years."

L455: you mention root:shoot ratio but then talk about 200-1000 cmg-1, which is a density and not a ratio, please correct.

Thank you for your suggestion. We have used the term: "ratios of root length to shoot biomass" and have corrected the term throughout the text, as in the lines below:

Line 469: now as: "In terms of the ratios of root length to shoot biomass, our observations were in line with those reported in the same soil types for wheat in Cai et al., (2018)".

Line 472: now as: "Jorda et al., (2022) reported a wide range of ratios of root length to shoot biomass from 200 to 1000 cm g$^{-1}$ around flowering time of maize depending on the wild type and root hair mutant genotypes growing on either loamy or sandy soils."

Line 477: now as "Cai et al., (2018) observed much larger ratios of root length to shoot biomass in drought stressed plots than in irrigated plot in both soil types in winter wheat which indicated the alternation of sink: source relationships to cope with water stress."

Line 493: now as "The larger ratios of root length to shoot biomass in this F2P2 plot in 2018 as compared to F2P3 could be explained by the change of source: sink relations where more assimilates were devoted to root growth, even at a later growth stage."

Line 687-689 (in Conclusion): now as "Our results confirmed that root length and ratios of root length to shoot biomass were modulated by soil types and water treatment but less by seasonal evaporative demand. Increase ratio of root length to shoot biomass has been an important response of maize that allows plants to extract more water under drought stress that occurred rather in the silty soil but less in the stony soil due to the higher content of stony material."

L435-478: This part of the discussion is too long, repeats a number of result part, and then brings elements of literature in a rather scattered and unrelated way (for instance talking about the soil nutrient effects on root length). And finally, it does not give a tangible conclusion of what we should retain about these root length differences between soil texture and between water treatments.

Many thanks for your suggestion. We revised and shortened the text by removing the repetition of result part (Line 459-465) and the soil nutrient effects on root length (Line 481-488), and ratio of root to leaf area (Line 466-469). Also, we removed the comparison of wheat (from previous studies, line 466-470 and line 481-488) and maize (our study) to make the discussion more focused on maize under different soils. In total, **we removed 17 sentences**. As suggested, we added one short sentence at the end of this section to conclude the root length differences were due to soil texture and soil water availability.

The original L435-478 is now much shorter, as:

"Our root observations showed that soil type considerably affected root growth more than water treatment (Figure 2). Root growth was strongly inhibited by the stony soil where much lower root length was observed than in the silty soil, especially in the deeper soil layers. This was consistent with the findings reported in (Morandage et al., 2021) where a linear increase of stone content resulted in a linear decrease of rooting depth across all stone contents and developmental stages. Also, both simulations and observations indicated that rooting depth was increased to the presence of cracks in the lower minirhizontron facility (Morandage et al., 2021) which could explain the high root length between 40 and 120 cm soil depths which was observed in the silty soil in both years.

In terms of the ratios of root length to shoot biomass, Ordóñez et al., (2020) has reported much larger figures of for instance 880 cm $g^{-1}$ in different locations and under different N application rates in maize growing in the Midwest of US. Jorda et al., (2022) reported a wide range of ratios of root length to shoot biomass from 200 to 1000 cm $g^{-1}$ around flowering time of maize depending on the wild type and root hair mutant genotypes growing on either loamy or sandy soils. More roots and higher ratios of root length to shoot biomass were found in the sand than in the loam in both wild type and root hair mutant genotypes (Jorda et al., 2022; Vetterlein et al., 2022). Cai et al., (2018) observed much larger ratios of root length to shoot biomass in drought stressed plots than in irrigated plot in both soil types in winter wheat which indicated the alternation of sink: source relationships to cope with water stress. This study emphasized that more assimilates are used to promote root growth and extract more water under drought stress. However, this was not the case for the stony soil in our work where the drought stress was more pronounced, especially in 2018. A drop of soil water potential (Supplementary material 2b), thus effective soil water potential (Figure 8a) was substantial from 10[th] July 2018 toward the harvest in the rainfed plot in the silty soil (F2P2) which was consistent with the reduction of leaf water potential (Fig. 8b), leaf area (Supplementary material 3c), total dry matter (Supplementary material 3d), and crop height (Supplementary material 4b) as compared the irrigated plot (F2P3). This indicates a mild water stress in 2018 in the rainfed plots on the silty soil. The larger ratios of root length to shoot biomass in this F2P2 plot in 2018 as compared to F2P3 could be explained by the change of source: sink relations where more assimilates were devoted to root growth, even at a later growth stage. Moreover, the low stone content and soil cracks (Morandage et al., 2021) might favor root growth in the deeper soil layers which are close to the shallow soil water table in the rhizotrone facility with silty soil (Vanderborght et al., 2010). In conclusion, both soil texture and water conditions influenced the root growth, however, effects of the former on root length was more pronounced than the latter."

L481: as affected by?

Thank you. We corrected the heading 4.2.1. Now as: "Leaf water potential and stomatal conductance as affected by soil water conditions"

L484: replace 'lower' by 'less'

We revised the sentence as suggested. Now line 488 as: "Moreover, the low stone content and soil cracks (Morandage et al., 2021) might favor root growth in the deeper soil layers which are close to the shallow soil water table in the rhizotrone facility with silty soil (Vanderborght et al., 2010)."

L482-489: you comment your results, then earlier results from others. And these are different. But you leave the reader hanging here: what is your interpretation/explanation for that?

Thank you. We revised the text: specifically moving the first sentence to the end of paragraph then adding one sentence for the explanation of impacts of soil on leaf water potential and stomatal conductance for the previous sentence. This will help the reader to have a clear message. Now in line 494-505 as:

"In the previous work, Koehler et al., (2022) reported that maize stomata closed at lower negative leaf water potentials in sand than in loam growing under controlled environment. Cai et al., (2022b) investigated transpiration response of pot-grown maize in two contrasting soil textures (sand and loam) and exposed to two consecutive VPD levels (1.8 and 2.8 kPa). Transpiration rate decreased at less negative soil matric potential in sand than in loam at both VPD levels. In sand, high VPD generated a steeper drop in stomatal conductance with decreasing leaf water potential which indicated that the transpiration and stomatal responses depend on soil hydraulics. In our study, stomata closed earlier and at more negative soil and leaf water potentials in the stony soil than in the silty soil (see Fig. 4, 5, 6 and 7). The lower soil water holding capacity of the stony soil compared to the silty soil resulted in lower soil water potential and smaller total plant hydraulic conductance which in turn led to earlier stomatal closure and to more negative soil water potential in the stony soil. "

L482-522: Here also is a fairly long piece of complex discussion that would need to end with a kind of message. I guess the message is that while Welcker and other have show genotypic differences earlier, here you show the soil influence on minimum psi Leaf.

Many thanks for your comment and suggestion. As suggested, we add one short sentence to conclude this key message. Now line 538 as: "In conclusion, our results confirmed that the minimum $\psi_{leaf}$ not only depended on genotypic differences but also was influenced by soil types and soil hydraulic conductance."

L525: Stomatal conductance?

This was revised.

L528: replace 'estimate' by 'estimation'

This was done. Thanks.

L533-539: Here I think it would be better to stick to the units of mm h-1 MPa-1 and then convert the values in 10-5 h-1 from the citations into that mm h-1 MPa-1 unit which has been used in your graphs.

Thanks. We changed the unit in $10^{-5}$ $h^{-1}$ into mm $h^{-1}$ $MPa^{-1}$ throughout the text from L563-569.

L540-541: That sentence is not understandable

We removed the lines since we think it is also not relevant for the context.

L649: contrasting

It was revised.

L666: …. and decreased at more negative…

It was revised. Now as: "The stomata conductance was smaller and decreased at more negative leaf water potentials in stony soil than in silty soil."

L667: remove 'plant'

Many thanks. The redundant word was removed.

L668: replace 'that' by 'and'

This was done. See also comment from Referee#1.

**Community comments – Dr. Oliver Dilly**

I suggest to delete "different soil types" at least in the title since this is misleading with reference to soil classification systems.

Only two contrasting soils has been considered differing in soil texture/ stone content; little information on soil type has been given.

Thank you very much for your comment. There were several studies have described more in detailed soil information in our study. We based on description in soil texture and particle content based on the three studies from Weihermüller et al., (2007), Stadler et al. (2015), and Morandage et al., (2021). For the field in our study, the dominant soil is a Haplic Luvisol that contains a layer with clay accumulation (silty loam texture) (Weihermüller et al., 2007). The thickness of the silty loam layer varies strongly along the slope of the field. It is up to 3 m thick at the bottom of the slope and not present at the top. One rhizotron facility is located at the top (stony) and has a stone content of >60% that this was referred to Cambisol [Geoscientific data from Geological Service North Rhine-Westphanlia, Krefeld (2012) with FAO soil taxonomy – IUSS Working Group WRB, (2006), see Fig. 1 in Stadler et al., (2015) and Table 1 in Morandage et al., (2021)]. The second rhizotron facility is located at the bottom of slope. It has a thick layer with silty loam and a negligible stone content (<4%) which is described as "stagnic Luvisol" [Geoscientific data from Geological Service North Rhine-Westphanlia, Krefeld (2012) with FAO soil taxonomy – IUSS Working Group WRB, (2006), see Fig. 1 in Stadler et al., (2015) and Table 1 in Morandage et al., (2021)]. This is a reason that we used the different soil types. We listed those three references the MS for the readers to have extra information with regards of soil types.

References:

Morandage, S., J. Vanderborght, M. Zörner, G. Cai, D. Leitner, et al. 2021. Root architecture development in stony soils. Vadose Zo. J. (April): 1–17. doi: 10.1002/vzj2.20133.

Weihermüller, L., Huisman, J. A., Lambot, S., Herbst, M., & Vereecken, H. (2007). Mapping the spatial variation of soil water content at the field scale with different ground penetrating radar techniques. Journal of Hydrology, 340, 205–216. https://doi.org/10.1016/j.jhydrol.2007. 04.013

Geological Service North Rhine-Westphalia, 2012 Geological Service North Rhine-Westphalia-GD.NRW (2012), Soil Map of North Rhine-Westphalia 1:5000, Procedure LA003 'Aldenhoven' (1969/2008) and Procedure W9506 'Ellen, WSG' (1984/1996).

Stadler, A., Rudolph, S., Kupisch, M., Langensiepen, M., Van Der Kruk, J., & Ewert, F. (2015). Quantifying the effects of soil variability on crop growth using apparent soil electrical conductivity measurements. European Journal of Agronomy, 64, 8–20. https://doi.org/10.1016/j. eja.2014.12.004

---

## Author Response (AR3)

Dear Dr. Vico (handling editor) and Dr. Rammig (co-editor-in-chef)

Thank you very much for your handling and support for the evaluation process. Also, thank you for email and request of major revision of the manuscript. To check the changes of MS, please kindly see the version with track changes and clean/final version. We responded to point to point comment and question from the handling editor. **The line numbers in the below responses are referred to the track changes version of the MS.**

On behalf of the authors

Thuy Nguyen

**Major comments**

I see you have addressed most of the other comments received. The reviewers who had originally assessed your work were unavailable for a second review and we could not secure an additional opinion, despite many trials. I am thus providing extensive comments on some of the remaining issues

Thank you very much for your great effort to handle the submission as well as your evaluable and constructive comments to improve the MS.

How was the irrigation amount and timing chosen? I understand this was discussed in a previous publication, but it is such an important point for these results that it needs to be explained around L162.

We added here the text to explain the irrigation as line **157 to 164**. Now as: "The irrigation systems [T-Tape 520-20-500 drip lines (Wurzelwasser GbR, Müzenberg, Germany)] were installed parallel to the crop rows with 0.3 m intervals. A nearby weather station (approx. 100 m from the experiment) recorded every 10 minutes weather variables (global radiation, temperature, relative humidity, precipitation, and wind speed). In addition, the precipitation amount was manually collected by a plastic rain gauge next to each rhizotrone facility. The Penman-Monteith equation was employed to estimate reference evapotranspiration. Daily crop evapotranspiration was calculated based on the single crop coefficient and the reference evapotranspiration (Allen et al., 1998). Irrigation amounts were estimated as the weekly sum of the calculated crop evapotranspiration."

Reference:

Allen, R.G., Pereira, L.S., Raes, D., Smith, M., 1998. FAO Irrigation and Drainage Paper – Crop Evapotranspiration. FAO, Italy.

More importantly, I see missing a proper discussion of the meaning of the two water treatments for the plant water status. Reading the discussion, it seems the assumption is made that rainfed=water stress and irrigated=well watered, but this seems unjustified (would that mean that in any year all rainfed crops in the region are water stressed? Are conditions such that irrigation is widely used?). I thus see the need to start the discussion with one on indicators of plant water status in the experiment. Of particular relevance are the consequences on plant water status of using the same irrigation strategy in 2017 and 2018. Moreover, the fact that the irrigated treatment was not replicated should appear as a caveat also in the discussion.

Thank you for this comment. We added one paragraph to discuss about this from **line 641-666**. As now:

"This study investigated soil-water-plant relations, more specifically the interactions of the root and shoot growth processes and water fluxes under variations of soil water status and atmospheric demands. To the best of our knowledge, the comprehensive data collected from soil to root, plant, and atmosphere under field conditions in this work was unique. However, we acknowledged the lack of treatment replicates which was due to the complex and expensive construction of the rhizotrone facilities. We also acknowledged the small size of plots that did not allow the extensive destructive sampling (i.e. leaf area, biomass, or determination of leaf water potential etc.). Each rhizotrone site originally contained the irrigated, rainfed, and rain-out sheltered plots (Nguyen et al., 2022a; Cai et al., 2016). The overall aim of the experiments was to investigate the root and shoot responses and gas fluxes ($CO_2$ and $H_2O$) of wheat and maize to the variations of soil water and soil hydraulics. Note that the studies did not intend to investigate the impacts of similar irrigation strategies on plant water status among seasons (i.e. in 2017 and 2018) because the irrigation practices were less common in the regions. The collapse of manual rain-out shelters due to strong wind after the 2016 growing season resulted in only two water treatments (rainfed and irrigated). Based on experiences from the previous seasons (wheat), we argued that such combinations of two water treatments and two soil types, to some extent, could still create a wide range of soil water conditions for the maize trial. For instance, the "rainfed" treatment at the stony soil in the upper rhizotrone (F1P2) could lead to severe water stress compared to other treatments, especially in the summertime when the atmospheric evaporative demand is high. In fact, mild water stress was observed at the F1P2 around mid -June in 2017. In 2018, the sites were slightly modified to induce more severe water stress (Nguyen et al., 2022a). One rainfed plot with the stony soil had late sowing while one rainfed plot with the silty soil had the higher sowing density (data not shown in the study). Unprecedented weather (extremely hot and dry) in 2018 resulted in severe drought stress at the rainfed plots with the stony soil. To compare the effects of soil types and water treatments on crop, we presented here only data from two plots (rainfed and irrigated) for two soil types. In spite of the experimental limitations, the relative differences among the treatments, soil types, and seasons as well as measured dates were clearly illustrated which ultimately supported the overall aim of our study."

I also suggest that the discussion is extended to present some implications of this work. They are not just mechanistic model parameterization! Note that adding these important points of discussion is not in contrast to some of the comments received before. These are important and very related points, whereas the reviewers had suggested to remove somewhat tangential (albeit interesting) material.

Thanks for this comments. We extended the discussion via adding one paragraph to elaborate some implications of our work. As now in line **667-686**:

"The simultaneous measurements of atmospheric conditions, leaf water potential, and transpiration rates, coupled with measurements of root, stem and whole soil-plant hydraulic conductance, root architecture (root length), and soil water potential distribution illustrated the complex responses of the shoot and root growth and hydraulic conductance vulnerability to soil water availability. The different responses of crop processes to soil hydraulics and climatic conditions suggest further field investigations for other soil types, growing seasons, and water regimes. Future studies considering the effects of progressive soil drying or irrigation strategies on plant water status and crop growth at field conditions will be necessary. This is very relevant for those crop-growing regions that require irrigation. Our results show that the leaf water potential threshold can vary within the same genotype depending on soil types, climatic conditions, and

water management. Large variability of minimum leaf water potential has been reported for maize genotypes under greenhouse conditions (Welcker et al., 2011). Field studies are required concerning the stomatal functions, water relations, hydraulic vulnerability traits, and root: shoot responses, especially of different maize cultivars in responding to drought stress. This will suggest implications for selecting agronomic cultivars and traits under changing climates. Results from this study show that soil-crop models should focus not only on simulating stomatal regulations to capture the response to drought stress, but also require adequate representations of root and leaf growth and adjustments. The soil hydraulics strongly influenced soil water availability and crop growths. Regional applications of soil-crop models for simulating gas fluxes and crop growth processes and for estimating irrigation amounts must account for the environmental heterogeneity within the spatial simulation unit whereas the soil heterogeneity is the key variable."

The conclusion section contains still some discussion points. For the best readability, the conclusions should be streamlined to be just that – conclusions – and any discussion point should be moved to the appropriate place in the discussion section.

Many thanks for your suggestion. The conclusion was streamlined. The text of discussion (line **694-698**) was put in the discussion section 4.3 (now as to line **607 and 620**). The text from **708 to 727** was moved to the discussion, section 4.1 (now as in line **464-484**). The line **732 to 734** related to implication to soil-crop model was moved to the new discussion (about the implication of the study) (see above response, line **667-686**).

**Minor comments:**

L63 an not as

This was corrected.

L94: something is amiss in this sentence

The sentence was corrected.

L199 and elsewhere: emerge is a verb; you can use emergence or, alternatively, emergence

This word has been corrected.

L308: what does "almost similar" mean? Be specific, possibly using statistical indicators (confidence intervals and the like)

We revised the text for this paragraph with more specific number and variables, as now in line **323-327**:

"Predawn and midday leaf water potential were around -0.4 MPa and -1.6 MPa for all plots, respectively. Leaf transpiration rate was around 3.1 millimole m$^{-2}$ s$^{-1}$ for all water treatments and soil types at 12 AM. This indicated the recovery of plant after watering at the rainfed plot with stony soil (F1P2).

L413: remove considerably

This was removed as suggested.

L470: climatic, non climate

This was corrected.

Note also some additional editorial comments provided as private note.

**Additional private note (visible to authors and reviewers only):**

Please note that there is still some substantial (self) plagiarism, which should be avoided. Please revise the manuscript to avoid that, or else we will be forced to reject that.

Thank you again for this suggestion. We substantially revised text to avoid the self-plagiarism, especially in the materials and methods and other small parts of the main text. More specifically in those lines:

Line 63

157-174 for water application and irrigation

Line 188-190: soil water measurement

Line 216-220 for leaf gas measurement

Line 222-230: sap flow measurements

Line 577

On a side note, will you make your data publicly available? Or will the data be available upon request? If you plan any of these, do specify where the data will be or that the data can be requested. Your decision on this matter will not influence the editorial decision.

We added here to data sources and citations for the below and aboveground data, respectively that could be publicly accessed. Two relevant publications (below and aboveground data) were added in the reference lists.

Lärm, L., F.M. Bauer, N. Hermes, J. van der Kruk, H. Vereecken, J. Vanderborght, Nguyen TH et al. 2023. Multi-year belowground data of minirhizotron facilities in Selhausen. Scientific Data 10(1): 1–1
Nguyen, TH, G. Lopez, S.J. Seidel, L. Lärm, F.M. Bauer, et al. 2024b. Multi-year aboveground data of minirhizotron facilities in Selhausen. Scientific Data 11(1): 1–11.

**Polina Shvedko**
**Notification to the authors:**
It seems that a table is included as Supplementary material 10. If it is so, it should be re-labelled as Table S1 and the references in the manuscript text should be adjusted accordingly. In this case figures of the supplement must be numbered consecutively: Figure S1, Figure S2, etc.

Many thanks for this guidance. The Figures and Table in the Supplementary material were updated as suggested. The names of those Figures and Table in the main text were also changed.